# 🗃️ TABLET: A LARGE-SCALE DATASET FOR ROBUST VISUAL TABLE UNDERSTANDING

**Iñigo Alonso[1], Imanol Miranda[2], Eneko Agirre[2], Mirella Lapata[1]**

[1]School of Informatics, University of Edinburgh
`{ialonso, mlap}@ed.ac.uk`

[2]HiTZ Center - Ixa, University of the Basque Country UPV/EHU
`{imanol.miranda, eneko.agirre}@ehu.eus`

## ABSTRACT

While table understanding increasingly relies on pixel-only settings, current benchmarks predominantly use synthetic renderings that lack the complexity and visual diversity of real-world tables. Additionally, existing visual table understanding (VTU) datasets offer fixed examples with single visualizations and pre-defined instructions, providing no access to underlying serialized data for reformulation. We introduce TABLET, a large-scale VTU dataset with 4 million examples across 21 tasks, grounded in 2 million unique tables where 88% preserve original visualizations. To evaluate whether models are able to jointly reason over tabular and visual content, we also introduce VisualTableQA, a benchmark requiring both visual perception and table understanding. Fine-tuning vision-language models like Qwen2.5-VL-7B and Gemma 3-4B on TABLET improves performance on seen *and* unseen VTU tasks while increasing robustness on real-world table visualizations. By preserving original visualizations and maintaining example traceability in a unified large-scale collection, TABLET establishes a foundation for robust training and extensible evaluation of future VTU models.[1]

## 1 INTRODUCTION

The field of table understanding focuses on techniques for representing and interpreting tabular data to support a wide range of practical tasks such as question answering, summarization, and information extraction. Research in this area has traditionally represented tables as structured text, encoding their content and layout through linearized or graph-based representations (see Figure 1b; Herzig et al. 2020; Zhang et al. 2020; Liu et al. 2022). While this unimodal view remains effective in certain domains, many tables found in documents and webpages contain irregular structures, rely on visual formatting (e.g., merged cells, background colors, font variations), or embed multimodal elements such as images (see Figure 1a). Advances in Vision-Language Models (VLMs; Radford et al. 2021; Liu et al. 2023) have provided impetus for treating tables as images, eschewing the step of rendering them as text sequences (like Markdown or HTML). The conceptual simplicity of this approach, coupled with improved performance on several tabular tasks (Alonso et al., 2024; Zhou et al., 2025) has driven significant research interest (Zheng et al., 2024b; Su et al., 2024; Jiang et al., 2025) in *Visual Table Understanding* (also known as *Multimodal Table Understanding*). Visual representations of tables are not only merely convenient but in many cases necessary, particularly for VLM agents that interact with the world exclusively through pixels (e.g., on a screen) and must interpret tables directly in their visual form (Deng et al., 2023; Zheng et al., 2024a; Lu et al., 2024).

Despite the growing relevance of VTU, there are few resources that support training models *directly* on image-based representations of tables. Existing benchmarks like MMTab (Zheng et al., 2024b) consist of web tables (e.g., from Wikipedia which is a common source for many tabular datasets), that are serialized and subsequently rendered as synthetic images (see Figures 1b,c). These images do not preserve the original visual characteristics of the tables and do not reflect the diversity and

---

[1]Dataset, code, and other resources are available at `https://github.com/alonsoapp/TABLET`.

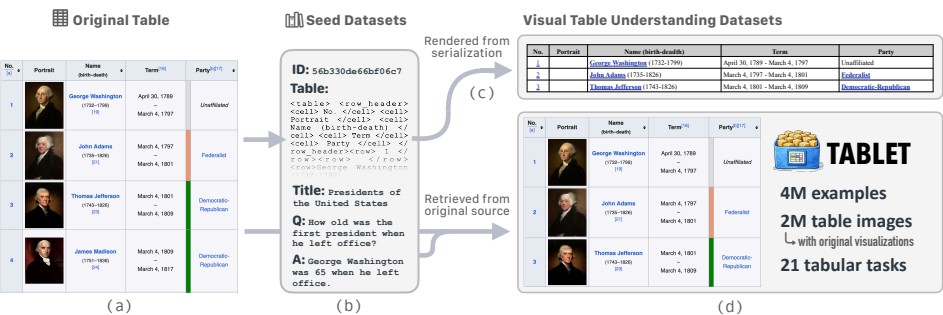

Figure 1: Previous datasets render table images from serialized tables, losing original visual details. In contrast, TABLET locates and retrieves the original table visualizations across 14 tabular datasets, resulting in 4M examples grounded in 2M unique tables.

complexity of real-world layouts. As a result, models trained on such data face a train-test mismatch, since the visual patterns learned from serialized renderings do not generalize well to naturally occurring tables failing to capture critical visual cues like subtle ruling lines, intricate merged cell layouts, background colors, font variations, or embedded images that are inherent to real-world table comprehension (compare Figure 1a and 1c). An exception is WikiDT Shi et al. (2024), which provides access to original table visualizations from web pages but focuses exclusively on table question answering. Likewise, datasets designed for processing PDF documents or screenshots (Zhong et al., 2020; Lu et al., 2023) preserve original table visualizations but support a single task.

In this work, we introduce TABLET[2], a large-scale dataset designed to enable vision-language models to learn generalizable skills for table understanding by leveraging visual features from table images. Unlike existing datasets that focus on a single task or use synthetic renderings, TABLET preserves *lossless representations* of real-world table images and spans a *diverse set of tasks*, including table-to-text generation (Parikh et al., 2020), table fact verification (Chen et al., 2020b), and table-based question answering (Pasupat & Liang, 2015). TABLET facilitates training on table formats that align with those encountered in downstream applications, particularly in pixel-based agents that process visual input directly. It contains 4 million examples across 21 tasks, derived from 2 million unique tables, 88% of which retain their original visualization. To promote flexibility and foster future research, we provide both image and HTML representations of tables, together with metadata and links to the source datasets. Pairing table images with HTML enables models to learn from naturally occurring and synthetic table images when the former are not available. TABLET's large scale and task diversity make it a foundational resource for research on integrating textual and visual modalities (Liu et al., 2025; Jiang et al., 2025).

To demonstrate TABLET's effectiveness as a training resource, we introduce VisualTableQA, a Visual Table Question Answering benchmark that pairs visually rich tables with questions that cannot be answered through table structure parsing or visual cues alone, but demand joint reasoning over both modalities. Our experiments demonstrate that fine-tuning on TABLET yields substantial improvements on VisualTableQA, as well as strong transfer to other in- and out-of-domain benchmarks. Our work makes the following contributions:

- We introduce TABLET to address a critical gap in VTU datasets, providing 4 million examples from 2 million tables, most preserved in their original visualization. We design TABLET for extensibility with a unified task format, HTML serializations that render back to original visualizations, and metadata with source links to facilitate reuse.
- Supervised fine-tuning on TABLET substantially improves VLM performance on both seen and unseen tasks. Our best-performing model achieves state-of-the-art results on 11 out of 15 table understanding benchmarks.
- We introduce VisualTableQA, a benchmark requiring joint table understanding and visual perception and show that fine-tuning on TABLET improves performance on this challenging unseen task.

---

[2]Our dataset is called TABLET partly as a nod to the Scottish sugary confection which is often cut into uniform squares or rectangles, resembling a table.

## 2    RELATED WORK

Early work approached the problem of table understanding from a unimodal perspective, treating tables as structured text and developing general-purpose models that generalized across multiple tasks (Li et al., 2023; Zhang et al., 2024). Recent progress in natural image understanding afforded by increasingly better VLMs (Radford et al., 2021; Liu et al., 2023) has led to development of models that can successfully process visually rendered text beyond natural imagery and perform table understanding holistically, within a multimodal framework (Kim et al., 2022; Lee et al., 2023; Zhang et al., 2023; Ye et al., 2023; Hu et al., 2024; Alonso et al., 2024; Jiang et al., 2025; Su et al., 2024; Zhao et al., 2024; Zheng et al., 2024b; Su et al., 2024).

To this effect, several benchmarks have been proposed recently to evaluate progress in VTU. For instance, TableVQA-Bench (Kim et al., 2024) focuses on visual table question answering but relies on synthetic table images and does not require visual reasoning: its questions can be answered from textual content alone. In contrast, MMTBench (Titiya et al., 2025) includes original table visualizations, along with visually rich images and interleaved charts. While both are valuable for evaluation, they are limited in scope and do not provide large-scale training data, restricting their utility for developing generalizable models.

Larger training datasets have historically focused on image-based Table Structure Recognition (TSR), including PubTabNet (Zhong et al., 2020), TableBank (Li et al., 2020), and TabComp (Gautam et al., 2025). Beyond TSR, Alonso et al. (2024) created image-rendered versions of existing datasets like ToTTo (Parikh et al., 2020), relying on lossy renderings that discard visual features. WikiDT (Shi et al., 2024) preserves original table visualizations but only targets question answering. MMTab (Zheng et al., 2024b) is a large VTU dataset, with 150k TSR pre-training examples and 232k instruction examples across 19 tasks. While MMTab is a useful resource that we build upon, it relies on synthetic renderings of serialized tables, lacks traceability to original data sources, and remains limited in size compared to the scale needed for general-purpose VTU.

In this work, we do not introduce a task-specific dataset or model. Instead, we create a large-scale resource aimed at enhancing table understanding in general-purpose vision-language models like Gemma-3 (Team et al., 2025) or Qwen2-VL (Wang et al., 2024), under the assumption that tables for many tasks are seen as images, and therefore can be naturally processed by these models. This distinguishes TABLET from many recent efforts that introduce task-specific datasets or specialized architectures as it is designed to be a holist resource for advancing VLM capabilities for tables.

## 3    THE TABLET DATASET

### 3.1    OVERVIEW

TABLET is a large-scale dataset that aggregates a total of 4,066,851 examples, combining 21 TU tasks sourced from 14 datasets (see Section 3.3 for a breakdown of examples per task). TABLET includes 2,031,256 unique table images with visualizations from Wikipedia (61.5%), PubTabNet (25.1%; Zhong et al. 2020), TabMWP (1.9%; Lu et al. 2023, and synthetically rendered images (11.5%) from serialized tables (see Section 3.2 for details).

Examples in TABLET are framed as instructions, leveraging the benefits of instruction-tuning for large language models. TABLET does not introduce new training data instances; rather it repurposes examples from existing datasets, referred to as *seed* datasets, rephrasing their tasks into an instruction format. Examples are drawn from 14 seed datasets: TURL (Deng et al., 2020), ToTTo (Parikh et al., 2020), TabFact (Chen et al., 2020a), WikiTableQuestions (Pasupat & Liang, 2015), HybridQA (Chen et al., 2020b), HiTab (Cheng et al., 2022), PubTabNet (Zhong et al., 2020), TabMWP (Lu et al., 2023), TAT-QA (Zhu et al., 2021), InfoTabs (Gupta et al., 2020), WikiBIO (Lebret et al., 2016), FeTaQA (Nan et al., 2022), MMTab Zheng et al. (2024b), and DocStruct4M (Hu et al., 2024). In addition, TABLET includes 306 newly created, human-annotated examples of questions based on visually rich tables drawn from six of the seed datasets.

Datasets that contain only images and fixed prompts restrict researchers to the provided visualizations and instruction formulations, limiting the exploration of new prompting techniques or the use of table data to create additional examples. To avoid this, each example in TABLET is provided in a

| Task | Seeds | Train | Dev | Test |
|---|---|---|---|---|
| ent_link | TURL | 1,236,128 (35.3%) | 74,282 (35.4%) | 213,494 (60.8%) |
| col_type | TURL | 602,406 (17.2%) | 13,188 (6.3%) | 12,802 (3.6%) |
| struct_aware_parse | M$^3$ | 513,482 (14.6%) | 9,115 (4.3%) | 1,102 (0.3%) |
| wikibio | WikiBIO | 582,659 (16.6%) | 72,831 (34.7%) | 72,831 (20.7%) |
| hybridqa | HybridQA | 62,670 (1.8%) | 3,466 (1.7%) | 3,463 (1.0%) |
| fetaqa | ToTTo | 3,006 (0.1%) | 577 (0.3%) | 1,079 (0.3%) |
| hitab | NSF, StatCan, ToTTo | 7,417 (0.2%) | 1,670 (0.8%) | 1,584 (0.5%) |
| infotabs | InfoTabs | 16,538 (0.5%) | 1,800 (0.9%) | 5,400 (1.5%) |
| tabfact | TabFact | 87,717 (2.5%) | 12,389 (5.9%) | 12,326 (3.5%) |
| tabmwp | TabMWP | 23,059 (0.7%) | 7,686 (3.7%) | 7,686 (2.2%) |
| tat-qa | TAT-QA | 2,201 (0.1%) | 278 (0.1%) | 277 (0.1%) |
| totto | ToTTo | 110,934 (3.2%) | 7,077 (3.4%) | 7,084 (2.0%) |
| wikitq | WikiTableQuestions | 14,152 (0.4%) | 3,537 (1.7%) | 4,344 (1.2%) |
| rel_extraction | TURL | 60,615 (1.7%) | 2,145 (1.0%) | 2,030 (0.6%) |
| table_instruction | M$^1$ | 136,944 (3.9%) | 0 (0.0%) | 0 (0.0%) |
| row_column_extraction | M$^1$ | 7,721 (0.2%) | 0 (0.0%) | 957 (0.3%) |
| table_cell_extraction | M$^1$ | 7,727 (0.2%) | 0 (0.0%) | 966 (0.3%) |
| table_cell_location | M$^1$ | 7,708 (0.2%) | 0 (0.0%) | 956 (0.3%) |
| table_recognition | M$^1$ | 6,927 (0.2%) | 0 (0.0%) | 912 (0.3%) |
| table_size_detection | M$^1$ | 7,800 (0.2%) | 0 (0.0%) | 950 (0.3%) |
| merged_cell_detection | M$^2$ | 7,500 (0.2%) | 0 (0.0%) | 950 (0.3%) |
| visual_table_qa | M$^4$ | 0 (0.0%) | 0 (0.0%) | 306 (0.1%) |
| Total | | 3,505,311 | 210,041 | 351,499 |

Table 1: TABLET splits, tasks, and seed datasets; M$^1$: InfoTabs, NSF, StatCan, TabMWP, TAT-QA, TabFact, ToTTo, WikiBIO, WikiTableQuestions; M$^2$: InfoTabs, NSF, StatCan, TAT-QA, TabFact, ToTTo, WikiBIO; M$^3$: PubTabNet, TabFact, WikiTableQuestions; M$^4$: ToTTo, HybridQA, TabFact, WikiBIO, TURL, WikiTableQuestions.

unified format, including the instruction used in this work, its corresponding atomic data, the HTML version of the table, the original source ID, and additional metadata (detailed in Appendix 7.4).

## 3.2 IMAGE SOURCES

**Wikipedia Tables** Wikipedia tables generally follow a hierarchical structure and are often rich in visual information. 61.5% of the tables in TABLET are lossless visualizations sourced from Wikipedia (see Appendix 7.2 for a breakdown of image sources per task). These tables are referenced in 67.4% of all examples in the dataset. Each table from the seed datasets was traced back to its original visualization in the corresponding Wikipedia article and captured in a screenshot.[3] A key challenge in retrieving the original visualizations was that the seed datasets were created at different times, while Wikipedia articles are continuously updated. To address this, we relied on the meta-data released with the seed datasets and also reached out to their authors to find out when the tables were harvested (see Appendix 7.3). We used Wikipedia's archiving API to retrieve each article as it appeared at the time of crawling. All Wikipedia tables in TABLET are linked to their source via the page identifier and the corresponding revision identifier (*oldid*). As Wikipedia pages often contain multiple tables, we identified which one corresponded to the seed example by computing the Levenshtein (1966) edit distance between the tables represented in markdown format, and selecting those with the highest similarity value. We set a minimum similarity threshold of 0.7. In cases without a match, the serialized seed table was rendered as an image.

**Synthetic Tables** Wikipedia tables that could not be retrieved, either due to low similarity scores or other issues, were synthetically generated by converting their serialized format to HTML and rendering them using the same browser configuration used for the original Wikipedia tables. These synthetic tables account for 11.5% of the dataset, contributing 747,002 additional examples (18.4% of the total). In Section 5, we show that combining original table images with synthetic visual-

---

[3]Table images were rendered using Firefox in headless mode (version 142.0.1 with GeckoDriver version 0.36.0) at an effective rendering density of 96 PPI (which stands for Pixels Per Inch).

izations during training leads to improved performance over using either alone, due to increased example count and visualization diversity.

**Document Tables** We also incorporate tables from other domains to improve model generalization. These include tables from scientific articles in PubTabNet (Zhong et al., 2020) and rendered tables from TabMWP (Lu et al., 2023). PubTabNet contributes 509,892 tables to our dataset (25.1% of the total), accounting for 97.5% of the structure-aware learning task incorporated from DocStruct4M (Hu et al., 2024). TabMWP adds 38,431 tables (1.9% of the total); while smaller in scale, TabMWP offers valuable coverage of numerical reasoning examples.

### 3.3    TABLE UNDERSTANDING TASKS

TABLET aggregates a total of 20 TU tasks grouped into 7 broad categories: Table Interpretation, Question Answering, Table-to-Text Generation, Table Numerical Reasoning, Table Natural Language Inference (NLI), and Table Structure Understanding. Subtasks within each category require different skills and demonstrate various degrees of complexity, ranging from basic table structure understanding to downstream tasks that combine tabular reasoning with information retrieval, numerical reasoning, or natural language inference. Table 1 provides a breakdown of examples per task and their source dataset. We briefly describe these below and provide more details in Appendix 7.5.

**Table Interpretation** comprises three tasks, namely *Column Type Annotation*, *Entity Linking*, and *Relation Extraction* (ent_link, col_type, and rel_extraction in Table 1). Sourced from TURL (Deng et al., 2020), these tasks do not represent downstream applications but target basic table understanding skills essential for tackling more complex tasks. All instructions for these tasks were aggregated from TURL. They constitute the largest group in our training set, with 1,899,149 examples (54.2%).

**Table Question Answering** is represented by *Free-form Table QA* (FeTaQA, Nan et al. 2022), *Hierarchical Table QA* (HiTabQA; Cheng et al. 2022), *Hybrid QA* (HybridQA; Chen et al. 2020b), and *Table QA* (WikiTableQuestions; Pasupat & Liang 2015). These tasks range from simple question answering over table content to multi-hop reasoning that combines textual and tabular information. They also vary in terms of table complexity and the expected answer format. Question answering represents 2.5% of TABLET's training set (87,245 examples).

**Table-to-Text Generation** is exemplified by two subtasks, namely *Highlighted Cell Text Generation* (ToTTo; Parikh et al. 2020) and *Wikipedia Biography Generation* (WikiBio; Lebret et al. 2016). Both tasks involve generating text based on table content, concise biographies in the case of WikiBio, and short descriptions based on visually highlighted cells, in the case of ToTTo. Together, they account for 693,593 training examples (19.8%).

**Table Numerical Reasoning** combines *Tabular Math Word Problem Solving* (TabMWP; Lu et al. 2023) and *Hybrid-context Financial QA* (TAT-QA; Zhu et al. 2021). Given a table and a mathematical question, the model must perform reasoning over table values. This type of numerical reasoning represents 0.8% of TABLET's training set (25,260 examples).

**Table NLI** includes two entailment tasks: *Table Fact Verification* (TabFact; Chen et al. 2020a) and *Infobox Natural Language Inference* (InfoTabs; Gupta et al. 2020) where statements as supported or refuted based on table content. They represent 104,255 examples (3%) in TABLET's training set.

**Table Structure Understanding** is an umbrella category for tasks designed to facilitate structural understanding of tables: *Merged Cell Detection*, *Row and Column Extraction*, *Table Cell Extraction*, *Table Cell Location*, *Table Recognition*, *Table Size Detection*, and *Structure-aware Parsing* in Table 1). Examples for the first seven tasks are aggregated from MMTab, while Structure-aware Parsing is derived from DocStruct4M (Hu et al., 2024). These tasks are based on tables from multiple seed datasets (see rows with seeds $M^2$, $M^1$, and $M^3$), including InfoTabs (Gupta et al., 2020), NSF (National Center for Science and Engineering Statistics, 2019), StatCan (Statistics Canada, 2024), TabMWP (Lu et al., 2023), TAT-QA (Zhu et al., 2021), TabFact (Chen et al., 2020a), ToTTo (Parikh et al., 2020), WikiBio (Lebret et al., 2016), WikiTableQuestions (Pasupat & Liang, 2015), and PubTabNet Zhong et al. (2020). While we maintain the original MMTab instructions, we use our own table visualizations. This group contributes 558,865 examples (15.8%) to our training.

**Instruction Following** To facilitate evaluation, we fine-tune and evaluate all models using instructions that explicitly require the final answer to be encapsulated in a JSON object, regardless of

any additional tokens the model may generate. To increase instruction diversity and mitigate catastrophic forgetting, we include a dedicated instruction-following set from MMTab. These instructions rephrase a subset of examples from the above tasks using a different template and do not require the JSON output format. While not a task in itself, this set adds instruction diversity and reduces overfitting to a single output style. We aggregate these examples directly from MMTab while using our visualizations. This set contributes 136,944 training examples (3.9%).

**Visual Table Question Answering**  Although many tasks in TABLET involve visually complex tables, they do not explicitly require models to exploit visual features beyond textual content. To evaluate whether training on TABLET's original images improves models' ability to integrate visual perception with table understanding, we introduce a manually curated Visual Table Question Answering benchmark. Human annotators selected tables with high visual complexity, as measured by our visual complexity scorer (Section 5 and Appendix 10), and formulated questions answerable only through joint reasoning over visual cues and tabular structure, e.g., *"What model is the red car in the table?"* or *"In what year did the king holding a red cross die?"*. To ensure questions genuinely require visual reasoning, we filtered out questions solvable from lossy synthetic representations alone. The resulting benchmark comprises 306 examples, each annotated with a question type to enable fine-grained performance analysis (see Appendix 13 for taxonomy and examples).

## 3.4 LINKING TO SOURCES AND HIGHLIGHTING

TABLET is designed with extensibility in mind. Existing datasets such as TableInstruct (Zhang et al., 2024), DocStruct4M (Hu et al., 2024), and MMTab (Zheng et al., 2024b) do not provide a clear reference to the original examples from which their instructions were derived. However, the absence of such pointers, makes reuse, modification, and augmentation difficult. Additionally, image-based datasets such as MMTab lack serialized table versions in text format, providing only rendered images as representations. To address these issues, each example in TABLET includes both the identifier of the original example and the identifier of the corresponding table from the source dataset. Identifiers follow a simple, human-readable format, and the released code provides a function to retrieve the original examples given their IDs. To support future extensions, TABLET also provides tables serialized in HTML. For tables retrieved from Wikipedia, this corresponds to the raw table object from the article's original HTML, which can be rendered using Wikipedia's official CSS stylesheets or the rendering functions included in our code. All remaining tables are either provided as HTML in the seed datasets or converted into HTML from their serialized forms.

This feature is particularly useful in tasks involving cell pointing or highlighting. Prior work has shown that models can achieve competitive results by relying uniquely on highlighted values (An et al., 2022; Alonso et al., 2024). We also observed in our experiments that as long as highlighted cell values are mentioned in the prompt, models are able to perform the task (e.g., Column Type Annotation, Entity Linking, Relation Extraction, FeTaQA, and ToTTo) regardless of the table provided. We mitigate this, by exploiting the HTML serialization to directly locate the highlighted cells in the source tables. By cross-referencing the highlighted spans from the seed datasets with the actual table content, we generate versions of the tables with explicit highlights while preserving the original visual features.

## 3.5 COMPARISON WITH MMTAB

MMTab (Zheng et al., 2024b) is also a dataset for visual table understanding; however, unlike TABLET, it does not include the full set of training examples from the original datasets (e.g., only 12.4% of ToTTo compared to our 91.8%; see Appendix 7.6), and it provides no development splits. MMTab covers 20 tasks while TABLET covers 22 (inculding instruction tuning); four tasks are unique to the former and five to the latter, and overall our dataset is 9.38 times larger (comprising 4,066,945 examples compared to MMTab's approximate 433,376 examples). Moreover, only a few tasks in MMTab include identifiers that can be linked back to their source examples, limiting developers to the provided instruction text. While MMTab relies on synthetically rendered tables with predefined styles, which omit original formatting and embedded images, TABLET retains the original visualizations, enabling models to exploit a fuller range of visual features.

| Model | WikiBio org | Δ | ToTTo org | Δ | WikiTQ org | Δ | HiTab org | Δ | TabFact org | Δ | FeTaQA org | Δ | Infotabs org | Δ | DegScore total |
|---|---|---|---|---|---|---|---|---|---|---|---|---|---|---|---|
| 0-Shot | 2.5 | +0.5 | 8.9 | -1.9 | 50.8 | -6.9 | 2.5 | -8.6 | 59.4 | -11.0 | 9.8 | -0.1 | **64.5** | -4.6 | -28.90 |
| TABLET-B$_{synth}$ | 4.8 | -6.5 | 14.4 | +2.0 | 56.4 | -1.9 | 16.0 | -17.5 | 65.5 | +2.4 | 27.9 | -0.7 | 51.8 | -0.6 | -22.35 |
| TABLET-B$_{org}$ | 11.5 | +1.9 | **15.9** | +2.2 | 56.2 | -2.3 | 24.7 | -10.4 | 63.7 | +1.6 | 28.4 | 0.0 | 48.7 | -2.7 | -6.63 |
| TABLET-B$_{mix}$ | **12.0** | +1.7 | 14.1 | +1.6 | **56.5** | -2.4 | **26.9** | -11.6 | **70.0** | +1.8 | **28.5** | -0.1 | 55.5 | -4.8 | -7.87 |

Table 2: Comparison of **Qwen2.5-VL-7B-Instruct** models fine-tuned on different TABLET-B variants and in zero-shot mode. We report performance when evaluating on original (lossless) table visualizations across held-in and held-out benchmarks and the corresponding degradation ($\Delta$) relative to evaluation on synthetic (lossy) tables. Negative $\Delta$ values indicate worse performance on original visualizations. We report exact match accuracy for WikiTQ, TabMWP, TabFact, and TAT-QA; BLEU for WikiBio, ToTTo, and FeTaQA; and F1 for HiTab (ToTTo on the dev set). Best results per setting (synth/org) are shown in bold. *DegScore* denotes the normalized aggregate degradation, computed separately for BLEU, accuracy, and F1 (see Appendix 8.2).

## 4 EXPERIMENTAL SETTING

Our experiments are motivated by three questions: (1) Does training with original visualizations lead to more robust generalization to real-world tables, even on benchmarks that do not explicitly require visual features, or does the added visual information act as noise and degrade performance?; (2) Does fine-tuning on TABLET outperform existing resources on both seen and unseen VTU tasks?; and (3) Does fine-tuning on TABLET enable models to better integrate visual perception and table understanding for unseen Visual Table Question Answering tasks? We also examine whether training only with original visualizations performs better than mixing original and synthetic ones, whether a balanced task distribution is preferable to the full dataset distribution, and whether including various table interpretation tasks contributes to VTU performance.

**Backbone Model** All main experiments employ the same backbone model, namely Qwen2.5-VL with 7B parameters (Qwen et al., 2025). This model provides a good tradeoff between performance and computational requirements. Together with InternVL3 (Zhu et al., 2025), it was the best-performing open-weight VLM in its class, setting a new benchmark for Single Page Document VQA (Mathew et al., 2021) at the time of writing. Exploring the best backbone model for VTU or advanced SFT strategies is outside the scope of this work. Full SFT configurations are detailed in Appendix 8; results for other open-weight models on TABLET (test set) are reported in Appendix 8.3.

**Evaluation** We evaluate fine-tuned models on eight *held-in* tasks: three focus on Table QA (WikiTableQuestions, HiTabQA, FeTaQA), two on text generation (WikiBio, ToTTo), two on Table Numerical Reasoning tasks (TabMWP, TAT-QA), and one on Table NLI (TabFact). To test generalization, we further evaluate on seven *held-out* tasks: VisualTableQA, the new benchmark introduced in this work, HybridQA, InfoTabs, TabMCQ (Jauhar et al., 2016), AIT-QA (Katsis et al., 2022), Table Recognition, and PubHealthTab (Akhtar et al., 2022). The last four are directly extracted from MMTab and use instructions generated with their own templates, adding diversity in both table and instruction styles. Although included in TABLET, the 86,135 training examples from HybridQA, InfoTabs, and Table Recognition were excluded from fine-tuning to enable held-out evaluation.

## 5 RESULTS AND ANALYSIS

**Lossless vs Synthetic Table Visualizations** We assess whether visual information present in lossless, real-world table visualizations improves model robustness. We conduct this experiment on a subset of TABLET corresponding to MMTab, the current state of the art in multimodal table understanding, which we refer to as TABLET-BASE (TABLET-B). We create three training variants: (1) TABLET-B$_{synth}$ uses only synthetic table renders (238,980 examples); (2) TABLET-B$_{org}$ replaces all Wikipedia-based images with original visualizations while keeping tables without lossy serialization unchanged (238,980 examples); and (3) TABLET-B$_{mix}$ extends TABLET-B$_{org}$ with synthetic renders for tables whose original visualizations could not be retrieved (371,292 examples; 43.5% original, 35.6% synthetic, 20.9% unchanged). We perform full fine-tuning (without LoRA) on

| | | Held-in Datasets | | | | | | | |
| | | WikiBio | ToTTo | WikiTQ | TabMWP | HiTab | TabFact | FeTaQA | TAT-QA |
|---|---|---|---|---|---|---|---|---|---|
| Qwen2.5-VL | 0-Shot | 2.5 | 9.1* | 53.4* | 59.1* | 31.2* | 73.9* | 7.0* | 6.9* |
| | 1-Shot | 4.4 | 16.1* | 49.8* | 57.7* | 37.3* | 72.8* | 8.5* | 10.1* |
| | MMTab | **6.4**\* | 12.6* | 48.7* | 80.4* | 41.5* | 57.0* | 1.7* | 9.4* |
| | TABLET-S | 2.9 | 28.9 | **56.8** | 84.0 | 64.8 | 78.9 | 28.7 | 27.8 |
| | TABLET-M | 3.1 | 29.3 | 56.6 | 84.0 | 67.0 | **79.5** | 30.7 | 31.0 |
| | TABLET-L | 3.8 | **30.4** | 55.5 | **84.5** | **67.5** | 79.5 | **31.5** | **32.5** |
| Gemma 3 | 0-Shot | 4.1 | 4.9 | **35.1** | 65.2 | 20.6 | **62.1** | 20.5 | 7.9 |
| | MMTab | 4.6 | 18.1 | 29.7 | **73.2**\* | 7.6 | 56.6 | 21.3 | 6.9 |
| | TABLET-M | 13.2 | **28.2**\* | 33.1 | 71.8 | **62.8**\* | 60.0 | **34.0**\* | **11.2** |
| | TABLET-L | **13.3**\* | 25.0 | 29.2 | 68.8 | 43.0 | 59.5 | 28.7 | 10.8 |

Table 3: Evaluation on eight held-in datasets with **Qwen2.5-VL-7B-Instruct** in 0/1-shot mode and fine-tuned on MMTab and different TABLET sizes. Results for Gemma 3 4B in 0-shot mode and fine-tuned on MMTab, TABLET-M, and TABLET-L using LoRA are also included. Results for ToTTo correspond to evaluation on the development set. We report exact match accuracy for WikiTQ, TabMWP, TabFact, and TAT-QA; BLEU for WikiBio, ToTTo, and FeTaQA; and F1 for HiTab. Best performing model per task is shown in bold; we mark with $*$ models significantly different ($p < 0.05$, using bootstrap resampling) from those trained on TABLET (highlighted in gray). Models that perform significantly better within the TABLET group are underlined. Results for other open-weight models on TABLET (test set) are reported in Appendix 8.3.

Qwen2.5-VL-7B for 3 epochs and evaluate on six held-in tasks and one held-out task, using both original and synthetic table images to simulate diverse training–evaluation conditions.

Table 2 shows that training on original visualizations substantially improves robustness to visual variation. The zero-shot baseline degrades by 28.9 percentage points when evaluated on original rather than synthetic tables, particularly on HiTab (-8.6) and TabFact (-11.0). Training on TABLET-B$_{synth}$ reduces this degradation to 22.35 points, but models still struggle with real-world visualizations. In contrast, TABLET-B$_{org}$ achieves the lowest degradation (6.63 points), demonstrating that exposure to original visualizations during training is crucial for robustness.

Comparing absolute performance on original tables, TABLET-B$_{org}$ outperforms TABLET-B$_{synth}$ on four tasks (WikiBio, ToTTo, TabFact, FeTaQA) and matches it on WikiTQ, underperforming only on HiTab and InfoTabs. The mixed training variant TABLET-B$_{mix}$ achieves the best overall results, winning on five of seven tasks (WikiBio, WikiTQ, HiTab, TabFact, FeTaQA) while maintaining low degradation (7.87 points). This demonstrates that combining original and synthetic visualizations leverages both the realism of original tables and the additional scale from synthetic renders, yielding models that are both accurate and robust across visualization styles.

**Performance on VTU Tasks with TABLET** We next evaluate whether TABLET improves performance on seen and unseen VTU tasks. While fine-tuned models are expected to perform better on tasks seen during training, improved performance on held-out datasets would suggest that training on TABLET enhances the model's VTU capabilities. Tables 3 and 4 report our results on held-in and held-out tasks, respectively. We present comparisons between zero- and one-shot Qwen2.5-VL-7B[4] and its fine-tuned instantiations trained on MMTab and different sizes of TABLET. For MMTab, all Wikipedia-sourced images are synthetic (61.8% of all tables in their dataset), whereas all TABLET versions use a mix of original (88%) and synthetic (12%) (See Appendix 7.2 for the distribution of image sources per task). For now, we focus solely on TABLET-L (a shorthand for TABLET-LARGE), which is our biggest dataset comprising 4M examples (we discuss smaller sizes in the next sections).

We observe that the model fine-tuned on TABLET-L performs significantly better on held-in *and* held-out datasets. It dominates in all QA benchmarks except HybridQA, where long-context profi-

---

[4]We should note that Qwen2.5-VL-7B is a competitive baseline that has undergone extensive pre-training and instruction-tuning, achieving top scores in many TU and Visual Document Understanding benchmarks.

|  | Held-out-Datasets | | | | | | |
|---|---|---|---|---|---|---|---|
|  | Infotabs | TabMCQ | AIT-QA | PubHealthTab | HybridQA | TabRec | VTQA |
| 0-Shot | **64.5**\* | 84.4 | 51.7\* | 64.7\* | **34.2**\* | 24.5\* | 42.4\* |
| MMTab | 56.9\* | **89.1** | 56.6\* | 63.9\* | 27.1 | 43.6\* | 41.1\* |
| TABLET-S | 57.2 | 88.2 | 58.9 | 64.7 | 22.8 | 43.9 | 45.2 |
| TABLET-M | 61.5 | 88.3 | 62.4 | 66.3 | 27.7 | 44.1 | **47.8** |
| TABLET-L | 61.4 | 87.9 | **70.8** | **70.2** | 25.1 | **45.4** | 45.6 |

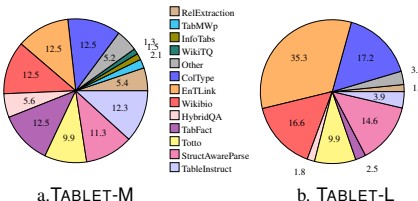

Table 4: **Left:** Evaluation on five held-out datasets with **Qwen2.5-VL-7B-Instruct** (0-shot) and fine-tuned on MMTab and different TABLET sizes. We report Tree-Edit- Distance-based Similarity for Table Recognition (TabRec) and accuracy for all other benchmarks. HybridQA results are reported on the development set. Best performing model per task is shown in bold; we mark with * models significantly different ($p < 0.05$, using bootstrap resampling) from those trained on TABLET (highlighted in gray). Models that perform significantly better within the TABLET group are underlined. **Right:** Distribution of examples per task in TABLET-L (b) and TABLET-M (a).

ciency is likely reduced since its training set was excluded from fine-tuning. Notably, in tasks such as HiTab, FeTaQA, and TAT-QA, where TABLET includes fewer training examples than MMTab, the model still performs considerably better, suggesting benefits from transfer across tasks, original visualizations, or simply due to a higher-quality subset of examples. Finally, tasks that include hierarchical table structures such as ToTTo and HiTab show clear gains. Table cell hierarchy is not lost in synthetic tables, but is often conveyed more clearly in their original visualizations. Our results are encouraging, particularly when considering that portions of the held-out datasets, including test sets, may have been encountered during pretraning. Models trained on TABLET outperform 0/1-shot Qwen2.5-VL-7B on 12 out 14 (held-in and held-out) tasks. Interestingly, our gains extend beyond unseen tasks. For instance, ToTTo, HiTab, TabFact, FeTaQA, and TAT-QA contain the *same* number of training examples across *all* TABLET variants, yet performance consistently improves as additional tasks are incorporated. This suggests that fine-tuning on a broader task set facilitates transfer learning, yielding improvements even on tasks where the model was already competitive.

**Optimal TABLET Size** Since training on 4M examples is resource-intensive, and tasks in TABLET-L exhibit substantial variation in size (see (b) pie chart in Table 4), we evaluate whether a smaller but more balanced dataset might perform comparably. We create TABLET-M (a shorthand for TABLET-MEDIUM) by capping each task at 140k examples. For benchmarks exceeding this cap (Column Type Annotation, Entity Linking, WikiBio), a random 140k subset is sampled and included as representative for that task. This results in a dataset with 1,117,217 training examples. We exclude HybridQA, InfoTabs, and Table Recognition from the training set to allow for held-out evaluation, leaving 1,031,082 examples in TABLET-M and 3,419,176 in the full dataset of our experiments. The capped distribution is shown in pie chart (b) and Table 4 (further details on training sets are in Appendix 7.1).

Models trained on TABLET-M perform competitively and in some cases, even better than TABLET-L (see Table 4, rows in gray), offering a compelling trade-off between dataset size and model effectiveness. That said, the full dataset still yields slightly better overall performance when fully fine-tuning a larger model and remains a valuable resource for future work that can benefit from large-scale, lossless training.

**Robustness Across Models and Training Regimes** We observe similar performance trends when training Gemma 3 4B with LoRA (Table 3, Appendix 12). Fine-tuning on TABLET-L yields substantial gains over the zero-shot baseline on six of eight tasks, with the largest improvements on visually rich datasets like HiTab (+22.4). Notably, both TABLET-M and TABLET-L significantly outperform MMTab on WikiBio, ToTTo, HiTab, and FeTaQA, despite using parameter-efficient LoRA rather than full fine-tuning employed for Qwen2.5-VL. This demonstrates that TABLET's benefits are robust to model architecture, model scale, and training methodology.

**The Benefit of Training on Table Interpretation Tasks** Table Interpretation tasks represent 54.2% of the training examples in TABLET-L, making them the largest category by volume and the second longest in instruction length, following HybridQA (see Appendix 8.1). While Deng et al. (2020) demonstrated the benefits of these tasks for textual TU, it is unclear whether this carries over to VTU. If these tasks prove unhelpful, removing them would save significant resources. We therefore remove all Table Interpretation tasks from TABLET-M and fine-tune on this smaller version (we use

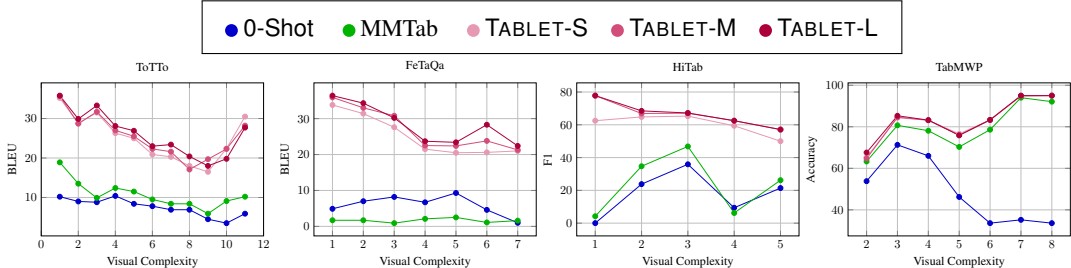

Figure 2: Model (**Qwen2.5-VL-7B-Instruct**) performance across different levels of table visual complexity (x axis). Higher levels correspond to more visually complex tables. Complexity levels are comparable across tasks, e.g., level 4 represents the same complexity range in every benchmark.

TABLET-S as a shorthand for TABLET-SMALL) which excludes any tasks related to Column Type Annotation, Entity Linking, and Relation Extraction resulting in a dataset with a total of 690,467 training examples (67% of the example count in TABLET-M). As shown in Table 4, there are benefits to be gained from including these tasks. Models trained on TABLET-M outperform those trained on TABLET-S on six out of 8 held-in benchmarks (Table 3) and achieve comparable performance on the remaining two. Similar gains are observed in four out of five held-out tasks (Table 4). These findings demonstrate that the benefits observed in the textual domain (Deng et al., 2020) extend to multimodal VTU.

**Performance on Unseen Visual Table Question Answering** Table 4 also reports results on VisualTableQA (VTQA), our benchmark designed to evaluate models' ability to jointly reason over visual cues and tabular structure. Fine-tuning on TABLET yields substantial improvements over both zero-shot (42.4%) and MMTab (41.1%) baselines, with TABLET-M achieving the best performance (47.8%). This represents a 13% relative improvement over zero-shot and 16% over MMTab, demonstrating that exposure to TABLET's visually rich original table images enables models to develop compositional skills that transfer to this challenging unseen task. Notably, these gains emerge without any task-specific VisualTableQA examples in the training data, confirming that TABLET's diverse task mixture and authentic visualizations foster generalizable visual reasoning capabilities. Results are averaged over three random seeds and reported as exact match accuracy.

**Visual Complexity Analysis** We hypothesize that TABLET is especially effective for visually complex tables; those deviating from regular, monochromatic layouts toward richer structures with stylistic cues, embedded images, irregular formatting, and diverse styling. These tables suffer most from serialization and motivate our work. Since TABLET contains both format-aware serialized representations (e.g., HTML) and rendered visual forms, we compute a visual-complexity score aggregating factors such as row/column-span irregularity, visual entropy, font diversity, embedded image features, and edge irregularity (see Appendix 10 for details). We rank tables by this score and group them into 20 equally spaced bins for analysis, omitting bins with fewer than 5 examples for clarity. Figure 2 shows results for tasks with the most pronounced model differences (see Appendix 11 for all benchmarks). Across these tasks, models trained on TABLET remain robust as visual complexity increases. For tasks like TABMWP, fine-tuning on table images proves crucial to avoid performance degradation, with TABLET-trained models showing consistently higher performance overall.

## 6 CONCLUSIONS

In this work, we introduced TABLET, a large-scale dataset for Visual Table Understanding that aggregates over 4 million examples across 21 tasks and 2 million unique tables. Unlike prior resources, TABLET preserves original table visualizations whenever possible, provides both HTML and image representations, and maintains full traceability to the source datasets. Through extensive experimentation, we showed that (1) training on TABLET improves model robustness when evaluated on real-world tables compared to synthetic renderings; (2) models fine-tuned on TABLET consistently outperform those trained on existing VTU datasets across both held-in and held-out tasks; and (3) TABLET brings improvements on unseen benchmarks, including the VisualTableQA benchmark introduced in this work, which suggests that its diversity supports transfer across VTU tasks.

ACKNOWLEDEGMENTS

We thank the anonymous reviewers for their feedback and Miao Li for useful discussions. We gratefully acknowledge the support of the UK Engineering and Physical Sciences Research Council (grant EP/W002876/1), the Ministry of Science, Innovation, and Universities of the Spanish Government MCIN/AEI/10.13039/501100011033 by means of the projects: (i) MOLVI (PID2024-157855OB-C32) and by FEDER, EU; (ii) HumanAIze (AIA2025-163322-C61), the CHIST-ERA grant (Project Geo-R2LLM, CHIST-ERA-23-MultiGIS-04) funded by the Ministry of Science, Innovation, and Universities of the Spanish Government (PCI2025-163286), the Basque Government (IXA excellence research group IT1570-22 and IKER-GAITU project), and the European Union under Horizon Europe (Project LUMINOUS, grant number 101135724).

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

## 7 APPENDIX

### 7.1 TABLET TRAINING SET STATISTICS

While the full released version of TABLET includes all available data, our experiments used different partitions. Specifically, to allow for held-out evaluation, we excluded HybridQA, InfoTabs, and Table Recognition from the training set; in addition, we created three training sets varying in size (TABLET-L, TABLET-M, TABLET-S) to explore various research questions. The statistics for these training sets are reported in Table 1. The development and test sets remain identical to the ones mentioned in the paper.

|  | Examples | Tasks | Original Images |
|---|---|---|---|
| TABLET-S | 690,467 | 14 | 521,032 (75.5%) |
| TABLET-M | 1,031,082 | 17 | 812,748 (78.8%) |
| TABLET-L | 3,419,176 | 17 | 2,795,485 (81.8%) |

Table 5: Training set statistics for TABLET-L, TABLET-M, and TABLET-S used to fine-tune Qwenn2.5-VL 7B in our experiments. We report the number of examples (Examples), the tasks used (Tasks), and the number of examples with original table visualizations.

### 7.2 IMAGE SOURCE PER TASK

Figure 3 shows the distribution of images per task included in TABLET. Images are sourced from Wikipedia, TabMWP, PubTabNet or are syntehtically rendered from information in the seed dataset.

### 7.3 DATASET CRAWLING DATES

We determined data collection dates for each dataset through various sources and methodologies. TURL originates from the TabEl dataset (Bhagavatula et al., 2015), which was crawled from the November 2013 English Wikipedia dump. For ToTTo, we obtained the collection date through direct correspondence with Ankur Parikh. TabFact's timeline was established based on email correspondence with Wenhu Chen and the initial arXiv publication date.

For HybridQA, we conducted a comprehensive analysis of all dates present within the dataset tables and observed a significant decrease in data frequency from February 2020 onward, indicating the collection cutoff point. InfoTabs' crawling year is documented at the bottom of their GitHub repository page, though the specific day represents our estimation. WikiBio's GitHub documentation indicates that their tables are sourced from the Common Crawl snapshot `enwiki-20150901`.

| Dataset | Date |
|---|---|
| TURL | 2013-11-01 |
| ToTTo | 2019-03-01 |
| TabFact | 2019-06-30 |
| HybridQA | 2020-01-31 |
| Infotabs | 2019-10-10 |
| WikiBio | 2015-09-01 |

Table 6: Data collection timeline for benchmarks included in TABLET.

### 7.4 DATASET EXAMPLE

For reference, in Figure 4 we show an example from TABLET for the ToTTo table-to-text task. Note that examples for every other task follow the same format. We provide the instructions used in our experiments, together with the corresponding example metadata, and links to the corresponding example and table from the source dataset.

### 7.5 DEFINITION OF TASKS REPRESENTED IN TABLET

As mentioned in Section 3.3, TABLET includes 21 TU tasks. We provide more detailed descriptions for each below.

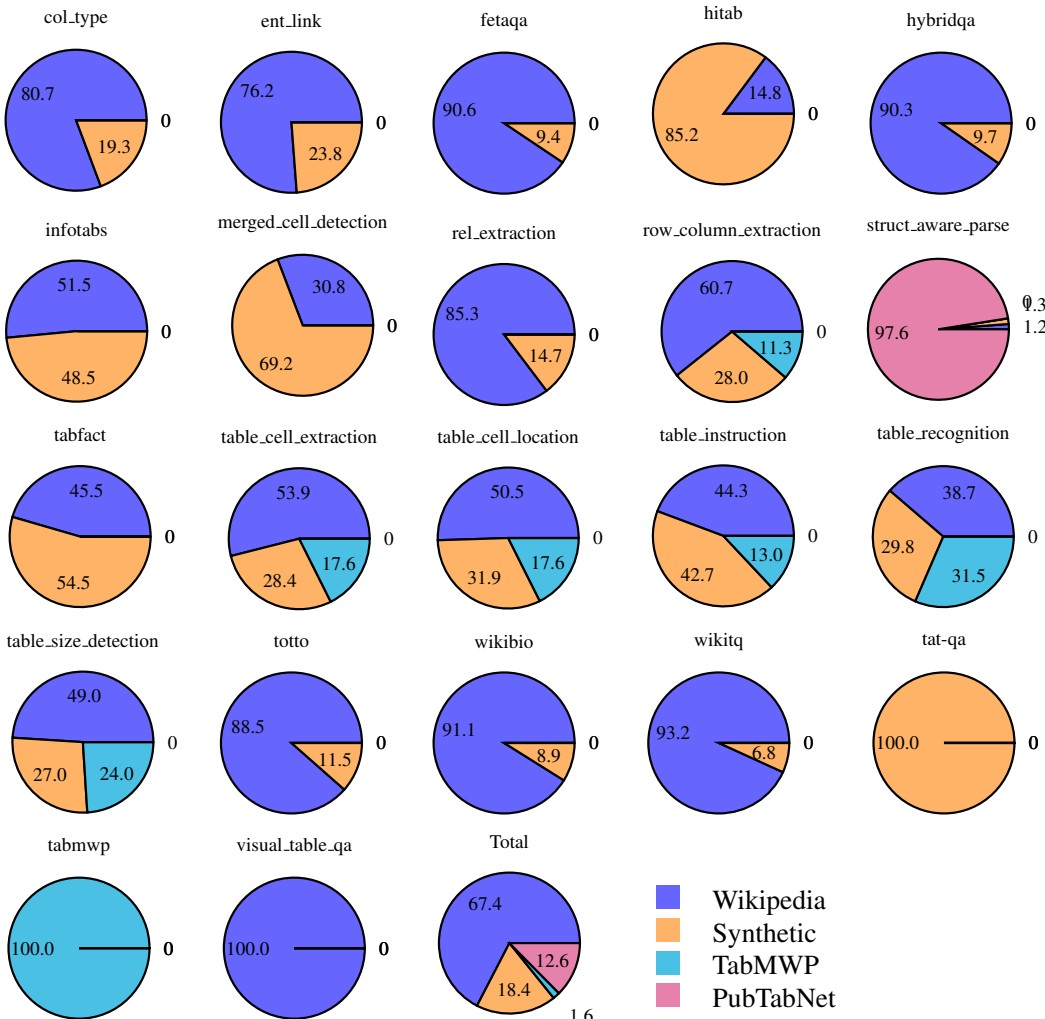

Figure 3: Image source distribution in TABLET, broken down by task. That is, source of the image referred by each example in each task. While the distribution resembles that of the unique image pool, it is computed at the example level (e.g., if the same Wikipedia image appears in two examples of a task, it is counted twice). Wikipedia are original visualizations form Wikipedia. Seed render are synthetic images rendered form information in the seed dataset.

**Column Type Annotation** In this task, the model is required to identify the data type of the values in a highlighted table column. The model selects from a set of 255 randomly shuffled candidate data types. There can be multiple correct types per column. The column is visually highlighted to ensure that the model attends to the table itself rather than relying solely on the textual instruction to infer the answer. Examples for this task were obtained from TURL. Our dataset includes 602,406/13,188/12,802 (train/dev/test) instructions for this task. An example of this task is shown in Figure 18.

**Entity Linking** For this task, the model is given a table with a visually highlighted cell along with a set of up to 100 candidate entities and descriptions and must identify which entity and description corresponds to the entity in the selected cell. This task was aggregated from the TURL dataset. Our dataset includes 1,236,128/74,282/213,494 (train/dev/test) instructions for this task. An example of this task is shown in Figure 6.

**Relation Extraction** This task requires the model to select appropriate relations between two visually highlighted columns of a table from a set of candidates. This task is derived from the

```
{
    "example_id": "e0af904e6617d0ab211ae76e62f7308c",
    "table_html_path": "html/highlighted/ToTTo/totto/train/254d984c233855f5be6814c6d1388c92_e0af904e6617d0ab211ae76e62f7308c.html",
    "table_img_path": "img/highlighted/ToTTo/totto/train/254d984c233855f5be6814c6d1388c92_e0af904e6617d0ab211ae76e62f7308c.png",
    "table_raw_html_path": "html/raw/ToTTo/totto/254d984c233855f5be6814c6d1388c92.html",
    "table_img_raw_path": "img/raw/ToTTo/totto/train/254d984c233855f5be6814c6d1388c92.png",
    "img_source": "wikipedia",
    "input": "Taking into account the table information on 'List of 8/9 PM telenovelas of Rede Globo', section '2000s', produce a
single-sentence summary focused on the highlighted cells and format it within the following JSON object {\"answer\": \"YOUR
ANSWER\"}.",
    "output": "{\"answer\": \"A Favorita is the telenovela aired in the 9 pm timeslot.\"}",
    "metadata": {"final_sentence": {"idx": [11, 69], "in": "output"}, "table_page_title": {...}},
    "table_seed_id": "1762238357686640028:train:0",
    "split": "train",
    "src_example_ids": {
        "ToTTo": "1762238357686640028:train:0"
    },
    "table_id": "254d984c233855f5be6814c6d1388c92",
    "table_page_title": "List of 8/9 PM telenovelas of Rede Globo",
    "table_section_title": "2000s",
    "table_seed_dataset": "ToTTo",
    "table_variant": "highlighted",
    "task": "totto",
    "table_wiki_page_id": "45544626",
    "table_wiki_old_id": 876845524,
}
```

Figure 4: Example of a TABLET example for the ToTTo table-to-text task.

TURL dataset. Our dataset includes 60,615/2,145/2,030 (train/dev/test) instructions for this task. An example of this task is shown in Figure 8.

**Structure Aware Parsing**    In this TSR task, the model needs to parse the table into markdown format. This task comes from Docstruct4M, using tables from PubTabNet, TabFact, and WikiTable-Questions. Our dataset includes 513,482/9,115/1,102 (train/dev/test) instructions for this task. An example of this task is shown in Figure 10.

**Free-form Table Question Answering (FeTaQA)**    In this task, the model generates free-form answers to questions about Wikipedia tables, often requiring integration of information from discontinuous sections of the table. Unlike datasets with shorter text spans, FeTaQA emphasizes higher-level understanding through long-form answers. Examples for this task come from FeTaQA dataset. Our dataset contains 3,006/577/1,079 (train/dev/test) instructions for this task. An example of this task is shown in Figure 7.

**Hierarchical Table QA (HiTabQA)**    This question-answering task involves hierarchical tables (with different headers across the table) and sometimes includes numerical reasoning, such as sums, averages, maximum, minimum, and counting, among others. Examples for this task were obtained from HiTabQA, with tables from ToTTo, StatCan, and NSF seed datasets. Our dataset contains 7,417/1,670/1,584 (train/dev/test) examples. An example of this task is shown in Figure 9.

**Table Fact Verification (TabFact, Infotabs)**    Also known as Table Entailment, this task involves classifying statements as supported or refuted based on table content. Examples for these two tasks come from TabFact and Infotabs seed datasets. Our dataset includes 87,717/ 12,389 / 12,326 (train / dev / test) TabFact examples and 16,538 / 1,800 / 5,400 (train / dev / test) Infotabs examples. An example of one of these tasks is shown in Figure 12.

**Table Numerical Reasoning (TabMWP, TAT-QA)**    Given a table and a mathematical question, the model must answer using mathematical reasoning over table values. Instructions were based on examples from TabMWP and TAT-QA. Our dataset includes 23,059 / 7,686 / 7,686 for TABMWP and 2,201 / 278 / 277 (train / dev / test) examples. An example of one of these tasks is shown in Figure 11.

**Table-to-Text (ToTTo)**    In this task, the model needs to generate a description based on the visually highlighted cells in a given table. Instructions were generated based on examples from ToTTo. Not all examples from the original dataset were retrieved, as we could not trace all highlighted cells from the dataset to the retrieved table for 8.1% of the examples. These examples were discarded to avoid adding noise to the dataset. Our dataset contains 110,934/7,077/7,084 (train/dev/test) examples. An example of this task is shown in Figure 13.

**Hybrid QA (HybridQA)**   This multi-hop question-answering task requires integrating structured table data with unstructured hyperlinked passages. Given a Wikipedia table and texts linked to the table's entities, the model must answer multi-hop questions by reasoning across both modalities. Instructions were generated using HybridQA's examples, resulting in a dataset containing 62,670/3,466/3,463 (train/dev/test) examples. An example of this task is shown in Figure 15.

**Table QA (WikiTableQuestions)**   Given a Wikipedia table and a question, the model must answer based on the table's content. For this task, WikiTableQA's examples are phrased as instructions in our dataset. Our dataset includes 14,152/3,537/4,344 (train/dev/test) instructions. An example of this task is shown in Figure 17.

**Wikipedia Biography Generation (WikiBio)**   Given a Wikipedia infobox of an entity, the model is prompted to generate a concise biography of this entity using the information in the infobox. This task's examples are aggregated from WikiBio. Our dataset includes 582,659/72,831/72,831 (train/dev/test) instructions. An example of this task is shown in Figure 14.

**Visual Table Question Answering**   Given a Wikipedia table and a question, the model must answer based on the table's content. Questions in this task require models to exploit visual features beyond textual content integrating visual perception with table understanding. This task's examples are original to this work and were produced by human annotators (See 3.3). Our dataset includes 0/0/306 (train/dev/test) instructions. An example of this task is shown in Figure 19.

**MMTab's Structure Understanding tasks**   TABLET includes all tasks from MMTab that are meant to instill table structure understanding in the model. These include: Merged Cell Detection, Row & Column Extraction, Table Cell Extraction, Table Cell Location, Table Recognition, Table Size Detection, and instruction following pre-training tasks. We refer to Zheng et al. (2024b)'s work for a description of these tasks. Instructions for these tasks are directly aggregated from MMTab and tables come from the following seed datasets: InfoTabs, NSF, StatCan, TabMWP, TAT-QA, TabFact, ToTTo, WikiBIO, WikiTableQuestions. TABLET maintains the same instructions as in MMTab but uses our visualizations for the table images. As instructions originate from MMTab, and no dev set is provided in this dataset, TABLET does not include any example for these tasks in the development set. Our dataset includes the following examples: Merged Cell Detection (7,500/0/950), Row & Column Extraction (7,721/0/957), Table Cell Extraction (7,727/0/966), Table Cell Location (7,708/0/956), Table Recognition (6,927/0/843), Table Size Detection (7,800/0/950), and instruction following pre-training tasks (136,944/0/0). An example of one of these tasks is shown in Figure 18.

### 7.6   MMTAB VS TABLET

Table 7 provides a detailed comparison between MMTab and TABLET across tasks and example counts (in training, development and test sets).

## 8   SUPERVISED FINE-TUNING DETAILS

All supervised fine-tuning (SFT) experiments were carried out with the same hyperparameters across models; the only varying factor was the dataset used for training. We fine-tuned **Qwen2.5-VL-7B-Instruct** using their official implementation.[5] Our code will be released alongside the dataset in their project repository.

**Hyperparameters**   All runs used the following common configuration:

- Training setup: DeepSpeed ZeRO-3, bf16 precision
- Epochs: 3
- Batch size: 2 (per device), gradient accumulation steps: 4 (for 32 GPUs) 8 (for 16 GPUs)
- Optimizer: AdamW, learning rate: 2e-7, weight decay: 0.01

---

[5] https://github.com/QwenLM/Qwen2.5-VL/tree/main/qwen-vl-finetune

| Task | Train | | Dev | | Test | |
|---|---|---|---|---|---|---|
| | MMTab | TABLET | MMTab | TABLET | MMTab | TABLET |
| wikibio | 4,994 | 582,659 | 0 | 72,831 | 1,000 | 72,831 |
| wikitq | 17,689 | 14,152 | 0 | 3,537 | 4,344 | 4,344 |
| totto | 15,000 | 110,934 | 0 | 7,077 | 7,700 | 7,084 |
| tabmwp | 30,745 | 23,059 | 0 | 7,686 | 7,686 | 7,686 |
| tabfact | 31,321 | 87,717 | 0 | 12,389 | 6,845 | 12,326 |
| hitab | 11,941 | 7,417 | 0 | 1,670 | 3,160 | 1,584 |
| infotabs | 18,338 | 16,538 | 0 | 1,800 | 5,400 | 5,400 |
| fetaqa | 8,327 | 3,006 | 0 | 577 | 2,003 | 1,079 |
| tat-qa | 5,920 | 2,201 | 0 | 278 | 772 | 277 |
| table_instruction | 37,204 | 136,944 | 0 | 0 | 0 | 0 |
| table_cell_extraction | 8,000 | 7,727 | 0 | 0 | 1,000 | 966 |
| table_cell_location | 8,000 | 7,708 | 0 | 0 | 1,000 | 956 |
| table_size_detection | 8,000 | 7,800 | 0 | 0 | 1,000 | 950 |
| merged_cell_detection | 8,000 | 7,500 | 0 | 0 | 1,000 | 950 |
| row_column_extraction | 8,000 | 7,721 | 0 | 0 | 1,000 | 957 |
| table_recognition | 8,000 | 6,927 | 0 | 0 | 1,000 | 912 |
| rotowire | 3,400 | 0 | 0 | 0 | 334 | 0 |
| col_type | 0 | 602,406 | 0 | 13,188 | 0 | 12,802 |
| ent_link | 0 | 1,236,128 | 0 | 74,282 | 0 | 213,494 |
| rel_extraction | 0 | 60,615 | 0 | 2,145 | 0 | 2,030 |
| hybridqa | 0 | 62,670 | 0 | 3,466 | 0 | 3,463 |
| struct_aware_parse | 0 | 513,482 | 0 | 9,115 | 0 | 1,102 |
| OOD | 0 | 0 | 0 | 0 | 1,250 | 0 |
| TabMCQ | 0 | 0 | 0 | 0 | 1,029 | 0 |
| AIT-QA | 0 | 0 | 0 | 0 | 511 | 0 |
| PubHealthTab | 0 | 0 | 0 | 0 | 1,942 | 0 |
| VisualTableQA | 0 | 0 | 0 | 0 | 0 | 306 |
| All | 232,879 | 3,505,311 | 0 | 210,041 | 49,976 | 351,499 |

Table 7: Comparison of MMTab and TABLET: tasks and examples across training, development, and test splits.

- Scheduler: cosine decay with warmup ratio 0.03

- Gradient clipping: 1.0

- Sequence length: 8192 tokens

- Vision input size: max_pixels = 50,176, min_pixels = 784

- Other: `data_flatten = False, data_packing = False, tune_mm_vision = False, tune_mm_mlp = True, tune_mm_llm = True`

**Compute Usage**    Table 8 reports the total GPU hours consumed by each experiment. All runs were conducted on clusters equipped with NVIDIA A100 GPUs.

## 8.1    INSTRUCTION SEQUENCE LENGTH

Figure 5 shows how instruction length (measured in terms of tokens) varies across tasks in TABLET. As can be seen, HybridQA has the longest instructions, followed by Column Type, Entity Linking, and Relation Extraction.

| Dataset | Setup | GPU Hours |
|---|---|---|
| TABLET-$B_{org}$ | 15h × 16 GPUs | 240 |
| TABLET-$B_{synth}$ | 15h × 16 GPUs | 240 |
| TABLET-$B_{mix}$ | 21h × 16 GPUs | 336 |
| MMTab | 21h × 16 GPUs | 336 |
| TABLET-S | 28h × 32 GPUs | 896 |
| TABLET-M | 40h × 32 GPUs | 1280 |
| TABLET-L | 125h × 32 GPUs | 4000 |
| Inference (TABLET-$B_{org}$) | 21h × 8 models | 168 |
| Inference (TABLET-$B_{synth}$) | 21h × 8 models | 168 |
| Inference (TABLET-$B_{mix}$) | 24h × 8 models | 192 |
| Inference (MMTab) | 24h × 8 models | 192 |
| Inference (TABLET-$B_L$) | 34h × 8 models | 272 |

Table 8: GPU hours for supervised fine-tuning and prediction runs. For the predictions, the 8 models are the 7 SFT models and the baseline model.

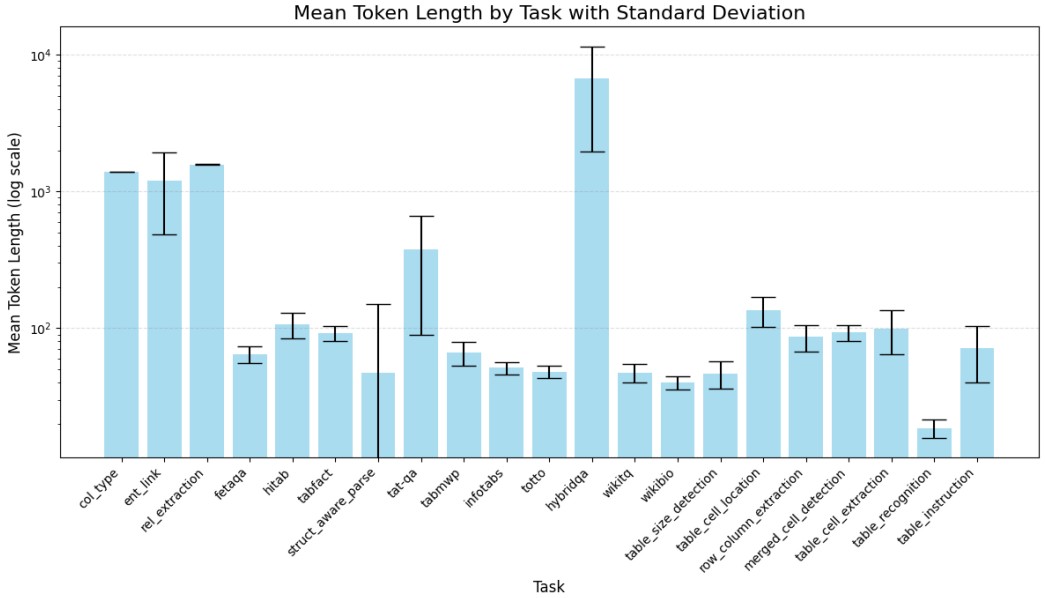

Figure 5: Instruction length distribution across tasks in TABLET.

## 8.2 PERFORMANCE DEGRADATION: ORIGINAL VS SYNTHETIC IMAGES

Let each task $t$ belong to one of the metric families $\mathcal{B}$ (BLEU), $\mathcal{A}$ (accuracy), or $\mathcal{F}$ (F1). For a given model $m$, we compute the raw difference:

$$\Delta_{m,t} = \text{Score}_{m,t}^{\text{org}} - \text{Score}_{m,t}^{\text{synth}}$$

Negative values indicate that the model performs worse when evaluated on the original visualizations compared to the synthetic ones. Because the metrics differ in scale, we normalize each by the corresponding synthetic score to report results in terms of percentage change:

$$\delta_{m,t} = \frac{\Delta_{m,t}}{\text{Score}_{m,t}^{\text{synth}}} \times 100.$$

Next, for each metric family $F \in \mathcal{B}, \mathcal{A}, \mathcal{F}$, we compute the family mean ($M$). Finally, the Degradation Score for model $m$ is defined as the unweighted average of the family means:

$$\text{DegScore} = \frac{1}{3}\Big(M_{m,\mathcal{B}} + M_{m,\mathcal{A}} + M_{m,\mathcal{F}}\Big).$$

This guarantees that BLEU, accuracy, and F1 contribute equally, regardless of how many tasks are included in each family.

### 8.3 OPEN-WEIGHT MODEL EVALUATION ON TABLET

Finally, in Tables 9 and 10, we report results on TABLET for open-weight VLMs other than Qwen2.5-VL-7B, in the zero-shot setting.

| | WikiBio | ToTTo | WikiTQ | TabMWP | HiTab | TabFact | FeTaQA | TAT-QA |
|---|---|---|---|---|---|---|---|---|
| Table-LLaVA 7B (Zheng et al., 2024b) | 5.7 | **16.3** | 17.7 | 47.5 | 10.6 | 53.6 | 7.5 | 1.8 |
| Qwen2.5-VL 7B (Qwen et al., 2025) | 2.5 | 9.1 | **53.4** | 59.1 | 31.2 | 73.9 | 7.0 | 6.9 |
| DocOwl2 8B (Hu et al., 2025) | 1.8 | 1.2 | 18.9 | 0.5 | 9.5 | 54.3 | 9.6 | 2.9 |
| InternVL3 8B (Zhu et al., 2025) | 4.6 | 11.9 | 45.3 | 83.9 | 40.8 | 64.3 | 3.8 | **13.0** |
| InternVL3 14B (Zhu et al., 2025) | **6.3** | 12.4 | 51.6 | **86.8** | **52.0** | **75.1** | 6.6 | 11.9 |
| Gemma-3 12B (Team et al., 2025) | 5.1 | 5.4 | 39.1 | 72.3 | 24.2 | 69.5 | **20.3** | 2.5 |

Table 9: Open-weight VLMs evaluated on TABLET held-in datasets (zero-shot setting).

| | Infotabs | TabMCQ | AIT-QA | PubHealthTab | HybridQA | TabRec | VTQA |
|---|---|---|---|---|---|---|---|
| Table-LLaVA 7B (Zheng et al., 2024b) | 61.5 | 59.1 | 4.7 | 50.2 | — | — | — |
| Qwen2.5-VL 7B (Qwen et al., 2025) | 64.5 | 84.4 | 51.7 | 64.7 | 34.2 | **24.5** | 42.4 |
| DocOwl2 8B Hu et al. (2025) | 15.5 | 63.0 | 36.8 | 14.6 | — | 4.6 | 9.5 |
| InternVL3 8B (Zhu et al., 2025) | 59.6 | 88.5 | **52.4** | 64.2 | 33.1 | 13.6 | 43.0 |
| InternVL3 14B (Zhu et al., 2025) | **67.4** | 88.8 | 52.3 | **74.9** | **46.7** | 13.4 | **49.9** |
| Gemma-3 12B (Team et al., 2025) | 65.5 | **88.9** | 48.7 | 64.7 | 35.2 | 7.1 | 28.5 |

Table 10: Open-weight VLMs evaluated on TABLET held-out datasets (zero-shot setting). Results for HybridQA are not reported for Table-LLaVA 7B and DocOwl2 8B due to their VRAM requirements, which do not scale well to the long contexts needed for our dataset.

## 9 LIMITATIONS

While TABLET is the largest resource for Visual Table Understanding to date, it has several limitations. Firstly, some original visualizations could not be retrieved due to missing or changed Wikipedia pages, and some embedded resources (e.g., images) were inaccessible. Secondly, we did not conduct a full ablation of the contribution of each task due to computational constraints, focusing instead on broader questions such as task balancing and dataset inclusion. Thirdly, most seed datasets are from Wikipedia, which, despite differing in visual format, are common in pretraining corpora, raising the possibility of data contamination. Finally, our fine-tuning focused on a single model (Qwen2.5-VL 7B), with zero-shot evaluation on others; future work should explore a broader range of architectures, especially those with reasoning or self-reflection capabilities.

**Table:**

| Place | Athlete | Time | Qual. |
|---|---|---|---|
| 1 | 🇨🇦 Mel Brock (CAN) | 1:57.0 | QS |
| 2 | 🇺🇸 Ted Meredith (USA) | | QS |
| 3 | 🏴 John Victor (RSA) | | |
| — | 🇬🇧 Alan Patterson (GBR) | DNF | |

**Instruction:**

```
For the Wikipedia table from the
↪   article 'Athletics at the 1912
↪   Summer Olympics Men's 800 metres'
↪   see 'Heats', select the proper
↪   entity that matches the highlighted
↪   table value given the following
↪   possible entities (<name /
↪   description / type>): <Alan
↪   Patterson / Wikipedia
↪   disambiguation page / None>, <Alan
↪   Patterson / British athlete /
↪   owl#Thing>, <Alan Patterson / UK MP
↪   / Person>, [...]. Return only the
↪   identifier or name of the chosen
↪   entity as JSON: {"answer": "YOUR
↪   ANSWER"}.
```

**Expected output:**

```
{"answer": "<Alan Patterson / British
↪   athlete / owl#Thing>"}
```

Figure 6: Example for Entity Linking task based on highlighted table cell.

**Table:**

| Year ⇕ | Competition ⇕ | Venue ⇕ | Position ⇕ | Event ⇕ | Notes ⇕ |
|---|---|---|---|---|---|
| Representing 🇷🇺 Soviet Union | | | | | |
| 1976 | Olympic Games | Montreal, Canada | 3rd | 4 × 100 m relay | |
| 1978 | European Indoor Championships | Milan, Italy | 1st | 60 metres | |

**Instruction:**

```
According to the table titled Nikolay Kolesnikov
↪   (sprinter), section Achievements, write a short
↪   sentence answer to the question: How did Nikolay
↪   Kolesnikov do at the 1976 Olympics and at the 60
↪   metres at the 1978 European Indoor Championships?
↪   Output only the correct answer as {"answer": "YOUR
↪   ANSWER"}.
```

**Expected output:**

```
{"answer": "Nikolay Kolesnikov won a bronze medal at
↪   the 1976 Olympics and won the 60 metres at the
↪   1978 European Indoor Championships."}
```

Figure 7: Example for Free-form Table Question Answering task based on highlighted table cells (FeTaQA).

**Table:**

| Nationality | Player | Ranking* | Seeding |
|---|---|---|---|
| ISR | Dudi Sela | 56 | 1 |
| TPE | Lu Yen-hsun | 62 | 2 |
| USA | Bobby Reynolds | 93 | 3 |
| GER | Michael Berrer | 111 | 4 |
| BRA | Thiago Alves | 120 | 5 |
| GER | Benjamin Becker | 126 | 6 |
| FRA | Nicolas Mahut | 137 | 7 |
| RUS | Michail Elgin | 138 | 8 |

**Instruction:**

```
For the table found in the article '2009 israel
↪  open' (section 'seeds') on Wikipedia, determine
↪  which relation holds between the two
↪  highlighted columns among these relation
↪  candidates:
↪  base.wikipedia_infobox.video_game.developer,
↪  organization.organization.headquarters,
↪  award.award_nominated_work.award_nominations,
↪  award.award_nomination.award_nominee, [...].
↪  Provide the correct relation as JSON:
↪  {"answer": ["RELATION"]} – list multiple
↪  relations comma-separated if necessary.
```

**Expected output:**

```
{"answer":
↪  ["people.person.nationality"]}
```

Figure 8: Example for Relation Extraction task based on highlighted table columns.

**Table:**

| | | Practice summary | | | |
|---|---|---|---|---|---|
| Session | Day | Fastest lap | | | |
| | | No. | Driver | Team | Car | Time |
| Practice 1 | Friday | 9 | David Reynolds | Erebus Motorsport | Holden VF Commodore | 1:12.8697 |
| Practice 2 | Friday | 88 | Jamie Whincup | Triple Eight Race Engineering | Holden VF Commodore | 1:12.5530 |

**Instruction:**

```
Answer the given question as concisely as possible
↪  using information from the table. Answers must be
↪  within a JSON object e.g. {"answer": ["YOUR
↪  ANSWER"]}. List multiple answers with commas
↪  inside the list object. Table caption: 'practice'.
↪  Relying on the table, answer the following: What
↪  was the time of david reynolds in practice 1?
```

**Expected output:**

```
{"answer": ["1:12.8697"]}
```

Figure 9: Example for Hierarchical Table QA task (HiTabQA).

**Table:**

| Diagnosis | Cases | Indeterminate | Evaluable |
| --- | --- | --- | --- |
| Follicular | 34 | 6/34 (17.6%) | 28/34 (82.4%) |
| Mantle cell | 7 | 0/7 (0%) | 7/7 (100%) |
| Marginal Zone | 11 | 1/11 (9%) | 10/11 (91%) |
| Lymphoplasmacytic | 6 | 0/6 (0%) | 6/6 (100%) |
| Diffuse Large B Cell | 15 | 2/15 (13.3%) | 13/15 (86.7%) |
| CLL/SLL | 4 | 0/4 (0%) | 4/4 (100%) |
| Small B Cell NHL, NOS | 2 | 0/2 (0%) | 2/2 (100%) |
| Total | 79 | 9/79 (11.3%) | 70/79 (88.6%) |

**Instruction:**

```
Convert the illustration to Markdown syntax.
```

**Expected output:**

```
<md> | Diagnosis | Cases | Indeterminate | Evaluable |
 | --- | --- | --- | --- |
 | Follicular | 34 | 6/34 (17.6%) | 28/34 (82.4%) |
 | Mantle cell | 7 | 0/7 (0%) | 7/7 (100%) |
 | Marginal Zone | 11 | 1/11 (9%) | 10/11 (91%) |
 | Lymphoplasmacytic | 6 | 0/6 (0%) | 6/6 (100%) |
 | Diffuse Large B Cell | 15 | 2/15 (13.3%) | 13/15
↪   (86.7%) |
 | CLL/SLL | 4 | 0/4 (0%) | 4/4 (100%) |
 | Small B Cell NHL, NOS | 2 | 0/2 (0%) | 2/2 (100%) |
 | Total | 79 | 9/79 (11.3%) | 70/79 (88.6%) | </md>
```

Figure 10: Example for Structure Aware Parsing task.

**Table:**

**Plants per garden**

| Stem | Leaf |
| --- | --- |
| 3 | 3 3 3 5 5 |
| 4 | 6 |
| 5 | 4 5 7 8 |
| 6 | 7 8 |
| 7 | 2 3 7 9 |
| 8 | 6 8 9 |

**Instruction:**

```
The members of the local garden club tallied the
↪   number of plants in each person's garden. How many
↪   gardens have at least 47 plants?
Solve the above problem according to the table called
↪   'Plants per garden'. Include your reasoning, after
↪   which finish with the JSON {"answer": "YOUR
↪   ANSWER"}.
```

**Expected output:**

```
Find the row with stem 4. Count all the leaves greater
↪   than or equal to 7.

Count all the leaves in the rows with stems 5, 6, 7,
↪   and 8.

You counted 13 leaves, which are blue in the
↪   stem-and-leaf plots above. 13 gardens have at
↪   least 47 plants. Therefore {"answer": "13"}
```

Figure 11: Example for Table Numerical Reasoning task (TabMWP).

**Table:**

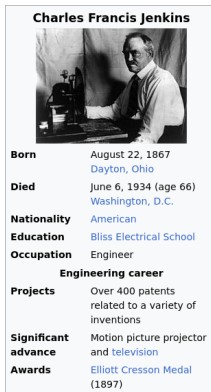

**Instruction:**

```
Consulting the table, state whether the hypothesis is
↪   entailed, neutral, or refuted. Give your final
↪   answer in JSON format {"answer": "YOUR ANSWER"}.

Hypothesis: Charles Francis Jenkins has been awarded
↪   more than one medal.
```

**Expected output:**

```
{"answer": "entailed"}
```

Figure 12: Example for Table Fact Verification task (Infotabs).

**Table:**

| Date | Time | | Score | | Set 1 | Set 2 | Set 3 | Set 4 | Set 5 | Total | Report |
|---|---|---|---|---|---|---|---|---|---|---|---|
| 20 Jul | 19:30 | **Slovakia** 🇸🇰 | **3–1** | Netherlands | 25–21 | 25–18 | 20–25 | 25–23 | | **95–87** | Report |

**Instruction:**

```
Consulting the following table concerning '2008 Men's
↪  European Volleyball League' in the 'Final'
↪  section, give a one-line statement limited to the
↪  highlighted cells and return it as JSON:
↪  {"answer": "YOUR ANSWER"}.
```

**Expected output:**

```
{"answer": "The 2008 Men's European Volleyball League
↪  was won by Slovakia, defeating the Netherlands by
↪  3{1 in the finals."}
```

Figure 13: Example for Table-to-Text task based on highlighted table cells (ToTTo).

**Table:**

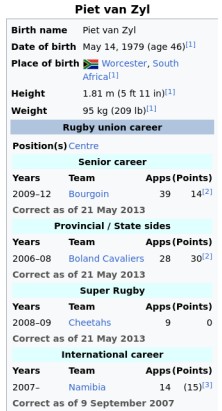

**Instruction:**

```
Drawing upon the provided infobox of piet van zyl
↪  -lrb- namibian rugby union player -rrb-, write a
↪  third-person, encyclopedic biography and format it
↪  within the following JSON object {"answer": "YOUR
↪  BIOGRAPHY"}.
```

**Expected output:**

```
{"answer": "Piet van zyl (born 14 may 1979) is a
↪  namibian rugby union player who captained the
↪  boland cavaliers in south africa at provincial
↪  level, and played for the at international level.
↪  Van zyl was in the namibian squad for the 2007
↪  world cup, and scored a try in his nation 's first
↪  match in the competition, in a game against. Van
↪  zyl plays as a centre. Van zyl made his debut in
↪  august 2007 in a friendly match against."}
```

Figure 14: Example for Wikipedia Biography Generation task (WikiBio).

**Table:**

| Rank ⬍ | Gymnasts ⬍ | Country ⬍ | Point ⬍ |
|---|---|---|---|
| ① | Attila Katus, Tamas Katus, Romeo Szentgyorgyi | 🇭🇺 Hungary | 16.55 |
| ② | Dorel Mois, Claudiu Moldovan, Claudiu Varlam | 🇷🇴 Romania | 16.25 |
| ③ | Maria Holmgren, Helene Nilsson, Kim Wickman | 🇸🇪 Sweden | 15.87 |
| 4 | Stanislav Marchenkov, Vadim Mikhailov, Denis Belikov | 🇷🇺 Russia | 15.55 |
| 5 | Grégory Alcan, Xavier Julien, Olivier Salvan | 🇫🇷 France | 15.00 |
| 6 | Won-Sil Choi, Hyun-Sung Ki, Kwang-Soo Park | 🇰🇷 South Korea | 14.95 |
| 7 | Marie-Catherine Boesa, Jana Heinze, Sandra Schlueter | 🇩🇪 Germany | 14.835 |
| 8 | Yumi Kobayashi, Kumi Sato, Hiroko Watabe | 🇯🇵 Japan | 13.758 |
| 9 | Giacomo Piccoli, Giovanna Lecis, Marco Bisciaio | 🇮🇹 Italy | 13.044 |

**Instruction:**

```
Taking into account the accompanying table as well as
↪  the context snippets provided at the end.

Answer the following question: What is the official
↪  name of the country that finished with the fifth
↪  most points at the 1998 Aerobic Gymnastics World
↪  Championships ? Respond with the correct answer
↪  (omit explanations) in JSON format as {"answer":
↪  "YOUR ANSWER"}. Do not introduce information
↪  beyond the provided sources.

Hungary is a country in [...]
Dorel Moi is a retired [...]
Claudiu Cristian Moldovan is a retired Romanian
↪  aerobic gymnast [...]
Claudiu Varlam is a retired Romanian aerobic [...]
Romania is a country located [...]
Sweden , officially the Kingdom of Sweden , is [...]
Russia , or the Russian Federation , is a [...]
France , officially the French Republic  , is [...]
South Korea , officially the Republic of Korea is
↪  [...]
Germany , constitutionally the Federal Republic of
↪  Germany , is [...]
Japan is an island country located in [...]
Italy , officially the Italian Republic , is [...]
```

**Expected output:**

```
{"answer": "French Republic"}
```

Figure 15: Example for Hybrid QA task (HybridQA).

**Table:**

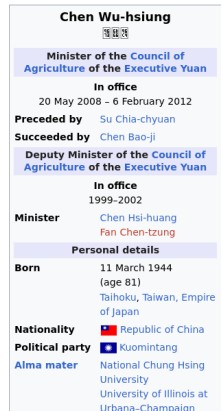

**Instruction:**

```
Based on the table image, extract the value of the
↪  cell located in the subsequent postion:
the 1st row and the 1st column

Format the cell value as a JSON, using the structure
↪  {"row_id":"m", "column_id":"n",
↪  "cell_value":"<Corresponding Cell Value>"}.
```

**Expected output:**

```
The target cell value in the 1st row and the 1st
↪  column is {"row_id":"1", "column_id":"1",
↪  "cell_value":"Chen wu-hsiung"}.
```

Figure 16: Example for Table Cell Extraction.

**Table:**

| Rank | City | Passengers | Ranking | Airline |
|---|---|---|---|---|
| 1 | 🇺🇸 United States, Los Angeles | 14,749 | — | Alaska Airlines |
| 2 | 🇺🇸 United States, Houston | 5,465 | — | United Express |
| 3 | 🇨🇦 Canada, Calgary | 3,761 | — | Air Transat, WestJet |
| 4 | 🇨🇦 Canada, Saskatoon | 2,282 | ▲ 4 | |
| 5 | 🇨🇦 Canada, Vancouver | 2,103 | — | Air Transat |
| 6 | 🇺🇸 United States, Phoenix | 1,829 | ▲ 1 | US Airways |
| 7 | 🇨🇦 Canada, Toronto | 1,202 | ▼ 1 | Air Transat, CanJet |
| 8 | 🇨🇦 Canada, Edmonton | 110 | | |
| 9 | 🇺🇸 United States, Oakland | 107 | | |

Busiest international routes at Manzanillo International Airport (2013)[1]

**Instruction:**

```
Answer this: How many more passengers flew to los
↪   angeles than to saskatoon from manzanillo airport
↪   in 2013? Consult the table and answer. Return your
↪   answer as JSON: {"answer": "YOUR ANSWER"}. Avoid
↪   including information not present in the table.
```

**Expected output:**

```
{"answer": "12,467"}
```

Figure 17: Example for Table QA task (WikiTableQuestions).

**Table:**

| Nationality | Player | Ranking* | Seeding |
|---|---|---|---|
| ISR | Dudi Sela | 56 | 1 |
| TPE | Lu Yen-hsun | 62 | 2 |
| USA | Bobby Reynolds | 93 | 3 |
| GER | Michael Berrer | 111 | 4 |
| BRA | Thiago Alves | 120 | 5 |
| GER | Benjamin Becker | 126 | 6 |
| FRA | Nicolas Mahut | 137 | 7 |
| RUS | Michail Elgin | 138 | 8 |

**Instruction:**

```
For the table found in '2009 israel open'
↪   - "seeds" section on Wikipedia,
↪   identify the correct column type
↪   labels for the highlighted column
↪   given the following type options:
↪   tv.tv_personality, time.event,
↪   american_football.football_team,
↪   [...]. Provide only the chosen
↪   type(s), separated by commas if
↪   multiple, within the list in this
↪   JSON: {"answer": ["ANSWER"]}.
```

**Expected output:**

```
{"answer": ["location.country",
↪   "location.location"]}
```

Figure 18: Example for Colum Type Annotation task based on highlighted table columns.

**Table:**

| Team | Points |
|------|--------|
| UST Tigresses | 31.0 |
| FEU Lady Tamaraws | 21.5 |
| UP Lady Maroons | 21.0 |
| Ateneo Lady Eagles | 13.5 |
| UE Amazons | 8.0 |
| NU Lady Bulldogs | 9.5 |
| De La Salle Lady Archers | Suspended |

**Instruction:**

```
Here is the question: Among the teams
↪   featuring blue in their colors, which
↪   one got the highest points? Using only
↪   the table, provide your answer. Return
↪   your answer as JSON: {"answer": "YOUR
↪   ANSWER"}. Avoid including information
↪   not present in the table.
```

**Expected output:**

```
{"answer": "Ateneo Lady Eagles"}
```

Figure 19: Example for Table Visual Question Answering task.

## 10 TABLE VISUAL COMPLEXITY METRIC

We create a visual complexity metric that assigns each table a score $S \in [0, 1]$, where 0 denotes a visually simple, regular, monochromatic table and 1 denotes a highly visually complex table. In practice, most complex tables in TABLET reach a score of 0.54 (see the distribution of scores in Appendix 11). The metric aggregates multiple structural and visual table characteristics: spanning irregularity, color and font diversity, image presence, entropy, and more, into a single normalized value via a weighted sum (weights listed in Table 11). The score combines structural HTML features with visual image analysis metrics.

### 10.1 HTML STRUCTURE METRICS

Let $N$ denote the total number of cells in the table. We extract the following structural features:

**Span Irregularity $S_{\text{span}}$**
Regular tables have cells occupying exactly one column and one row (colspan= 1, rowspan= 1). We measure irregularity through two components:

- **Proportion of irregular cells:** Let $n_{\text{col}}^{(1)}$ and $n_{\text{row}}^{(1)}$ denote the number of cells with colspan= 1 and rowspan= 1, respectively. The proportion irregularity is:

$$I_{\text{prop}}^{\text{col}} = 1 - \frac{n_{\text{col}}^{(1)}}{N}, \quad I_{\text{prop}}^{\text{row}} = 1 - \frac{n_{\text{row}}^{(1)}}{N} \tag{1}$$

- **Span magnitude:** Let $c_i$ and $r_i$ denote the colspan and rowspan values for cell $i$. The weighted span magnitude is:

$$I_{\text{mag}}^{\text{col}} = \frac{1}{N} \sum_{i=1}^{N} \frac{c_i - 1}{\max(c_i, 1)}, \quad I_{\text{mag}}^{\text{row}} = \frac{1}{N} \sum_{i=1}^{N} \frac{r_i - 1}{\max(r_i, 1)} \tag{2}$$

The final span irregularity combines both components with equal weight:

$$S_{\text{span}} = \frac{1}{2} \left[ \frac{1}{2}(I_{\text{prop}}^{\text{col}} + I_{\text{mag}}^{\text{col}}) + \frac{1}{2}(I_{\text{prop}}^{\text{row}} + I_{\text{mag}}^{\text{row}}) \right] \tag{3}$$

**Color Diversity $S_{\text{color}}$**
Let $n_{\text{bg}}$ and $n_{\text{text}}$ denote the number of unique background and text colors. Simple tables typically use 1-2 colors. We compute:

$$S_{\text{color}} = \frac{1}{2} \left[ \min\left(\frac{n_{\text{bg}} - 1}{5}, 1\right) + \min\left(\frac{n_{\text{text}} - 1}{5}, 1\right) \right] \tag{4}$$

**Font Diversity $S_{\text{font}}$**
Let $n_{\text{fonts}}$ denote the number of distinct font families. Simple tables use a single font:

$$S_{\text{font}} = \min\left(\frac{n_{\text{fonts}} - 1}{3}, 1\right) \tag{5}$$

**Image Presence $S_{\text{img}}$**
Let $n_{\text{img}}$ denote the number of embedded images. We normalize by table size:

$$S_{\text{img}} = \min\left(\frac{n_{\text{img}}}{0.3 \cdot N}, 1\right) \tag{6}$$

### 10.2 VISUAL IMAGE METRICS

Given a rendered table image $\mathbf{I} \in \mathbb{R}^{H \times W \times 3}$, we compute the following visual features:

**Visual Entropy $S_{\text{entropy}}$**
We convert the image to grayscale and compute the Shannon entropy of the intensity histogram. Let $p(i)$ denote the normalized frequency of intensity value $i \in \{0, \ldots, 255\}$:

$$S_{\text{entropy}} = \frac{-\sum_{i=0}^{255} p(i) \log p(i)}{\log 256} \tag{7}$$

High entropy indicates pixel intensity randomness, while low entropy suggests uniform regions.

**Color Complexity $S_{\text{cplx}}$**

Let $n_{\text{unique}}$ denote the number of unique RGB triplets. We normalize by a fraction of total pixels $P = H \times W$:

$$S_{\text{cplx}} = \min\left(\frac{n_{\text{unique}}}{0.1 \cdot P}, 1\right) \tag{8}$$

**Edge Irregularity $S_{\text{edge}}$**

We apply Sobel operators to detect horizontal ($\nabla_x$) and vertical ($\nabla_y$) edges. Regular grids exhibit balanced edge distributions:

$$S_{\text{edge}} = \min\left(\frac{\sigma(\{|\nabla_x|_{\text{mean}}, |\nabla_y|_{\text{mean}}\})}{50}, 1\right) \tag{9}$$

where $\sigma$ denotes standard deviation and $|\cdot|_{\text{mean}}$ the mean absolute gradient magnitude.

**Color Saturation $S_{\text{sat}}$**

Converting to HSV color space, let $\mathbf{S}_{\text{HSV}}$ denote the saturation channel. The saturation score is:

$$S_{\text{sat}} = \frac{1}{255 \cdot P} \sum_{h,w} \mathbf{S}_{\text{HSV}}(h, w) \tag{10}$$

**Non-White Background Ratio $S_{\text{bg}}$**

Let $\mathcal{P}_{\text{white}}$ denote pixels where all RGB channels exceed 240. The non-white ratio is:

$$S_{\text{bg}} = 1 - \frac{|\mathcal{P}_{\text{white}}|}{P} \tag{11}$$

### 10.3 FINAL COMPLEXITY SCORE

The overall complexity score is a weighted combination of all component scores:

$$S = \sum_k w_k \cdot S_k \tag{12}$$

where the weights $w_k$ sum to 1. We use the following default weights:

| Component | Weight ($w_k$) |
|---|---|
| Span Irregularity ($S_{\text{span}}$) | 0.12 |
| Color Diversity ($S_{\text{color}}$) | 0.15 |
| Font Diversity ($S_{\text{font}}$) | 0.08 |
| Image Presence ($S_{\text{img}}$) | 0.10 |
| Visual Entropy ($S_{\text{entropy}}$) | 0.15 |
| Color Complexity ($S_{\text{cplx}}$) | 0.12 |
| Edge Irregularity ($S_{\text{edge}}$) | 0.10 |
| Color Saturation ($S_{\text{sat}}$) | 0.10 |
| Non-White Background ($S_{\text{bg}}$) | 0.08 |

Table 11: Component weights for table complexity score computation.

The resulting score $S \in [0, 1]$ provides a continuous measure of table complexity. Figure 20 shows four tables with progressively increasing visual-complexity scores.

## 11 TABLE VISUAL COMPLEXITY ANALYSIS

We report the full visual complexity analysis across all benchmarks in Figure 21. For each task, we report model performance as a function of table visual complexity using the metric introduced in Appendix10.

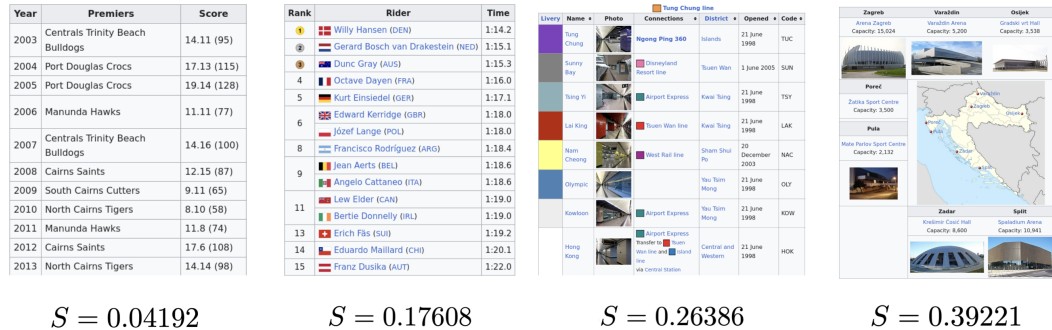

$$S = 0.04192 \qquad S = 0.17608 \qquad S = 0.26386 \qquad S = 0.39221$$

Figure 20: Example tables with increasing visual-complexity levels and their corresponding scores.

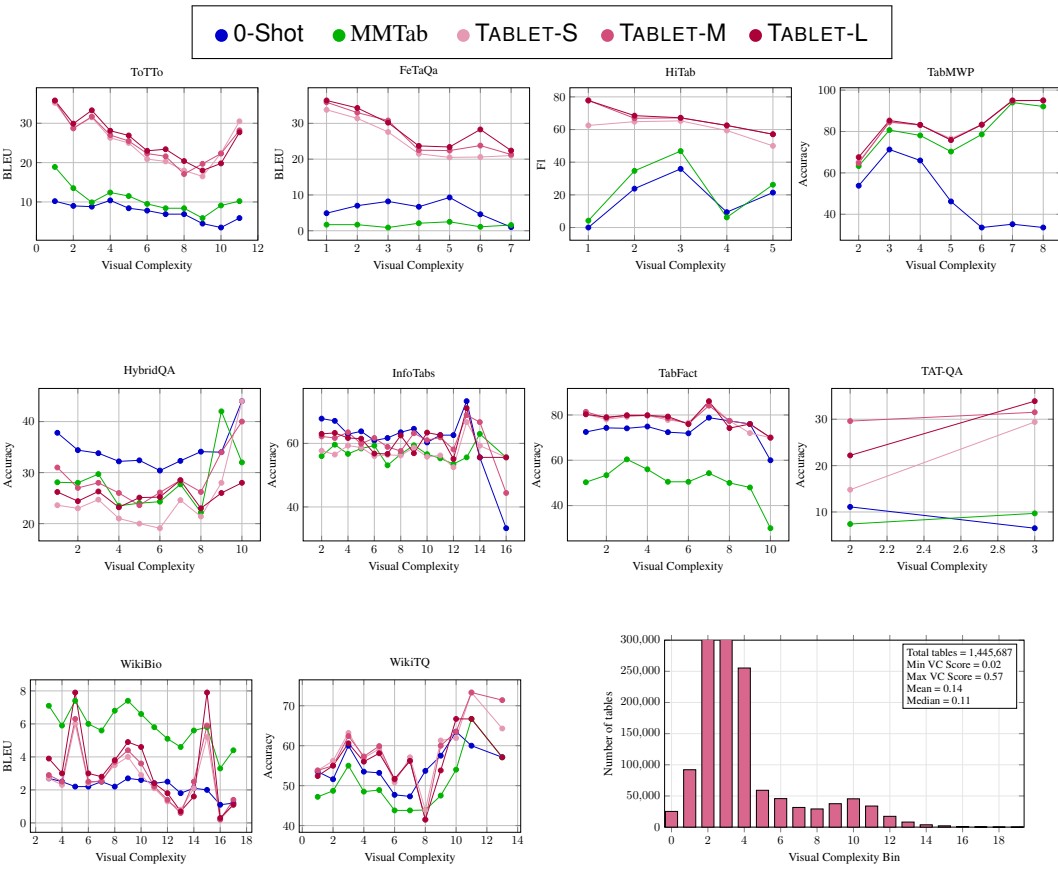

Figure 21: Model (**Qwen2.5-VL-7B-Instruct**) performance across levels of table visual complexity (x axis). Higher levels correspond to more visually complex tables. Complexity levels are comparable across tasks, that is, e.g., level 4 represents the same complexity range in every benchmark.

## 12 GEMMA 3 TRAINING SETUP

We fine-tuned the Google Gemma-3-4B-IT model using Low-Rank Adaptation (LoRA) on TABLET-M. In this section we disclose the training hyper-parameters.

- **Base Model**: `google/gemma-3-4b-it`
- **Quantization**: 4-bit NormalFloat (NF4) with double quantization

- **Compute dtype**: `bfloat16`
- **Attention Implementation**: Flash Attention 2
- **Maximum Sequence Length**: 8,192 tokens
- **LoRA Rank** ($r$): 16
- **LoRA Alpha** ($\alpha$): 16
- **LoRA Dropout**: 0.05
- **Target Modules**: All linear layers
- **Bias**: None
- **Additional Trainable Modules**: `lm_head`, `embed_tokens`
- **Number of Epochs**: 1
- **Per-Device Batch Size**: 1
- **Gradient Accumulation Steps**: 64
- **Effective Batch Size**: 256
- **Learning Rate**: $2 \times 10^{-4}$
- **Learning Rate Scheduler**: Constant
- **Optimizer**: AdamW (fused implementation)
- **Warmup Ratio**: 0.03
- **Maximum Gradient Norm**: 0.3
- **Precision**: bfloat16 mixed precision training

## 13 VISUALTABLEQA QUESTION CATEGORIES.

VisualTableQA examples can belong to one or many of the following categories:

- **V1. Image-based Attribute Identification:** Requires interpreting pixel-level image content within table cells (e.g., Example 22a).
- **V2. Visual Formatting-based Selection:** Relies on non-image visual cues such as text color, cell shading, or row styling (e.g., Example 22b).
- **S1. Structural / Positional Reasoning:** Involves reasoning over table layout or relative position (e.g., Example 22c).
- **R1. Aggregation and Numerical Reasoning:** Requires counting, aggregation, or identifying extrema (e.g., Example 22d).
- **R2a. Single-hop Conditional Retrieval:** Retrieves a value from a row identified by a single condition (e.g., Example 22e).
- **R2b. Multi-hop / Comparative Retrieval:** Requires multiple constraints, comparisons, or nested selection (e.g., Example 22f).
- **B1. Multiple Choice:** Questions formulated as binary or multiple-choice (e.g., Example 23a).

(a) **V1. Image-based Attribute Identification**

**Q:** Is Edmund J. Davis wearing a military uniform?
**A:** *yes*

(b) **V2. Visual Formatting-based Selection**

**Q:** Which team is in the gray row?
**A:** *Newbee*

(c) **S1. Structural / Positional Reasoning**

**Q:** Which team is directly above Australia in the table?
**A:** *England*

(d) **R1. Aggregation and Numerical Reasoning**

**Q:** How many different nationalities are shown in the table?
**A:** *2*

(e) **R2a. Single-hop Conditional Retrieval**

**Q:** Which country does the red house belong to?
**A:** *Bulgaria*

(f) **R2b. Multi-hop / Comparative Retrieval**

**Q:** Which are the movements for the entry that has the same background color as 2004 and 2005?
**A:** *Relegated*

Figure 22: Example questions from each VisualTableQA category (part 1 of 2).

(a) **B1. Multiple Choice**

| Common name ⬍ | Binomial name ⬍ | Population ⬍ | Status ⬍ | Trend ⬍ | Notes | Image |
|---|---|---|---|---|---|---|
| **Little Spotted Kiwi** | *Apteryx owenii* | **1,200**[1] | NT[1] | ⌐[1] | Minimum estimate. [1] | |
| **Northern Cassowary** | *Casuarius unappendiculatus* | **3,500 – 15,000**[2] | VU[2] | ▼[2] | | |
| **Great Spotted Kiwi** | *Apteryx haastii* | **8,000**[3] | VU[3] | ▼[3] | | |
| **Southern Cassowary** | *Casuarius casuarius* | **10,000 – 19,999**[4] | VU[4] | ▼[4] | | |
| **Southern Brown Kiwi** | *Apteryx australis* | **29,800**[5] | VU[5] | ▼[5] | | |
| **North Island Brown Kiwi** | *Apteryx mantelli* | **35,000**[6] | VU[6] | ▼[6] | Preliminary estimate. [6] | |

**Q:** What is the trend of the bird with the lowest population?
Answer: positive, negative, same, or unknown.
**A:** *same*

Figure 23: Example questions from each VisualTableQA category (part 2 of 2).

