# OpenReview forum: "TABLET: A Large-Scale Dataset for Robust Visual Table Understanding"
_ICLR.cc/2026/Conference — ICLR 2026 Poster_

### Official Review · Reviewer_qfkf · 2025-10-31

**Soundness:** 3
**Presentation:** 3
**Contribution:** 3
**Rating:** 6
**Confidence:** 3

**Summary:**

Current visual table understanding (VTU) benchmarks rely on synthetic data lacking real-world complexity and omit underlying serialized data, while TABLET is a large-scale dataset with 4 million examples from real visualizations, paired image-HTML representations, and 20 tasks. Fine-tuning vision-language models on TABLET boosts their performance on seen/unseen VTU tasks and robustness to real-world tables, establishing a foundation for future VTU model training and evaluation.

**Strengths:**

TABLET addresses prior single-task/synthetic data limitations with large-scale, lossless real-world table visualizations and 20 diverse tasks, while its high quality lies in 4M examples, image-HTML pairings, and traceable metadata. Its clear unified format and extensible design, coupled with significant contributions to boosting VLM cross-task performance (including SOTA results) and enabling practical VTU research, underscore its field value.

**Weaknesses:**

1. Whether all tables are presented in English? As far as I am aware, Wikipedia provides original HTML files in multiple languages.
2. Each example in TABLET is accompanied by the HTML version of the corresponding table. Can such a large volume of data be used to train table parsing models? Is the annotation format of HTML consistent? For instance, how are formulas within the tables annotated? Additionally, do these HTML files contain certain formatting information, such as row widths and font colors?

**Questions:**

see the weakness

---

> ### Author Response · Authors · 2025-11-26
>
> We thank the reviewer for their feedback, and especially for taking the time to read and review our paper. In the following response, we will address your concerns and comments point-by-point.
>
> > 1. Whether all tables are presented in English? As far as I am aware, Wikipedia provides original HTML files in multiple languages.
>
> Our dataset currently focuses on English-language tables, sourced from English Wikipedia and other English-language documents. We acknowledge that creating a fully multilingual version is non-trivial, as Wikipedia article content varies between languages and we would need to verify that there is a 1:1 match of table information across language versions.
>
> However, besides full traceability of the original example and Wikipedia article for Wikipedia sourced tables, TABLET also includes the HTML of each table that allows future researchers to translate and re-render tables in other languages with no information loss. This is something that would not be possible with datasets that lack this underlying serialization.
>
> > Each example in TABLET is accompanied by the HTML version of the corresponding table. Can such a large volume of data be used to train table parsing models? Is the annotation format of HTML consistent? For instance, how are formulas within the tables annotated?
>
> Yes, TABLET can indeed be used to train table parsing models. Each example includes an HTML version that is 1:1 matched with the corresponding table image. This characteristic can also be used for contrastive learning approaches.
>
> The HTML formatting of all non-wikipedia based tables is highly consistent. Wikipedia sources tables however, are generated by a programmatic translator from Wikipedia’s own marked language to HTML, so we can also expect a significant level of consistency across tables. For example, formulas are typically displayed under the tag <math>, having access to all HTMLs allows us to analyze table characteristics like these easily, for example, there are 107 tables displaying formulas in TABLET with a total of 322. This analytic would have been significantly difficult to compute from images alone.
>
> > Additionally, do these HTML files contain certain formatting information, such as row widths and font colors?
>
> Yes, in fact we leverage this information in the visual complexity analysis we have included in the paper during this rebuttal period. We have conducted an analysis on how different visual factors affect models performance and how models trained on our dataset behave under different levels of visual complexity. We included this analysis in the “Visual complexity analysis” paragraph in Section 5.
>
> To perform this analysis, we created a metric that aggregates multiple structural and visual table characteristics: column and row span irregularity, color and font diversity, embedded image presence, color entropy, and more, into a single normalized value (please find the full details of this metric in Appendix 10 of our new version of the paper). This metric is very effective at creating a scale of visual complexity across the 1.4M tables in our dataset (please see Figure 19 in the Appendix of the revised paper). This is another example of how creating such a metric would have been significantly more difficult without access to both a format-aware serialized representation (HTML) and the rendered visual form that we include in TABLET.
>
> We compute this score for all tables in TABLET and evaluate each model's performance within each group for each benchmark. Figure 2 in the revised paper shows results for the tasks where model differences are most pronounced (see Appendix 11 for results across all benchmarks).
>
> Across these tasks, models trained on our dataset remain robust even as tables deviate substantially from regular, visually simple layouts. The analysis also shows that in some tasks like TABMWP, fine-tuning on table images is crucial to avoid performance degradation, and TABLET trained models consistently show higher performance overall.
>
> We apologize for the rather late response as we have been busy working on applying the feedback we have gathered during the rebuttal. We hope we addressed the reviewer's concerns along with the value of the newly added analysis and other experiments.

---

### Official Review · Reviewer_WHyK · 2025-11-01

**Soundness:** 2
**Presentation:** 3
**Contribution:** 2
**Rating:** 6
**Confidence:** 4

**Summary:**

This paper identifies a critical "train-test mismatch" in the field of Visual Table Understanding (VTU). The authors argue that existing benchmarks predominantly use "synthetic renderings" of tables, which lack the visual complexity (e.g., merged cells, colors, fonts, embedded images) of real-world tables. This causes models trained on them to fail when generalizing to real-world visual data.

To solve this, the authors introduce TABLET, a new large-scale dataset of 4 million examples over 20 tasks. The dataset's primary contribution is that 88% of its 2 million unique tables are "original visualizations" meticulously retrieved from historical web snapshots (primarily Wikipedia), preserving their true visual fidelity. The authors demonstrate through experiments that while performance on traditional tasks is mixed, training on TABLET dramatically improves a model's robustness to visual domain shift (synthetic vs. original) and enhances generalization to unseen tasks.

**Strengths:**

1. **Problem Identification:** The paper's core premise is strong and well-articulated. It correctly identifies the "train-test mismatch" as a fundamental problem for the VTU field as it shifts towards pixel-only VLMs. This is a significant and practical issue.

2. **Methodological Rigor on Data Collection:** The primary contribution is the dataset's "visual fidelity." The engineering effort to achieve this is non-trivial and highly commendable. The authors detail a rigorous process of tracing seed datasets to their original crawl dates and using Wikipedia's archiving API to retrieve historical snapshots, followed by Levenshtein matching to find the correct table. This high-quality execution is a major strength.

3. **Strong Robustness Argument:** The paper's most compelling evidence is in Table 9. It clearly shows that models trained on synthetic data suffer a massive performance degradation (-22.35 points) when evaluated on original visualizations. In contrast, models trained on TABLET's original data are far more robust (only -6.63 points). This single finding strongly validates the paper's core hypothesis and the necessity of this dataset.

4. **Resource Value and Extensibility:** The paper delivers more than just a static benchmark; it provides a large-scale, multi-task resource. By including HTML representations, metadata, and traceability links to source datasets, the authors enable future research, task reformulation, and extensibility.

**Weaknesses:**

1. **Core Claim Undermined by Own Results:** The paper's most significant weakness is that its primary experiment (Table 2) fails to support its central claim. When comparing models trained on TABLET-B_org (original) vs. TABLET-B_synth (synthetic), the performance is roughly equivalent across most tasks. This result is underwhelming and directly contradicts the motivation that visual fidelity improves performance on these tasks.

2. **Circular Argument:** The authors' explanation for the failure of Table 2 is that the benchmarks themselves (e.g., ToTTo, WikiTQ) are flawed and were designed for text, not visual cues. This is a "catch-22" and a form of circular reasoning: the paper claims its dataset is the solution but simultaneously admits that the tasks it uses for validation are incapable of proving it. They have built a high-fidelity testbed but failed to provide tasks that actually require that fidelity.

3. **Diluted Conclusion on "Mixed" Data:** In several experiments (Table 2, Table 10), the TABLET-B_mix (original + synthetic) model performs best. This dilutes the paper's central premise. The takeaway risks becoming "more data is better" (a trivial conclusion) rather than the intended, stronger claim that "high-fidelity data is better."

4. **Limited "Real-World" Diversity:** The paper claims to solve the "real-world" visualization problem, but the dataset's diversity is questionable. 88% of the data (61.5% of tables) is sourced from Wikipedia. While better than synthetic data, Wikipedia tables are still highly structured and relatively uniform. This is not representative of the true "in-the-wild" chaos of scanned PDFs, financial reports, or product pages.

**Questions:**

1. Given that the core experiment in Table 2 fails to show a clear superiority for TABLET-B_org over TABLET-B_synth, and you attribute this to flawed benchmarks (e.g., ToTTo), does this not imply that the primary contribution of this work is a resource for which the tasks do not yet exist?

2. The strong performance of the TABLET-B_mix model seems to suggest that a combination of synthetic and original data is optimal. Does this not contradict the paper's central premise that synthetic renderings are inherently flawed and should be replaced by original visualizations?

3. Why were new, visually-demanding tasks (e.g., "find the cell with the red background," "what is the value in the merged cell?") not introduced and evaluated to actually prove the hypothesis that the visual fidelity of TABLET is its key advantage?

---

> ### Author Response · Authors · 2025-11-27
>
> We apologize for the rather late response, we sincerely appreciate the time and effort it took the reviewer to write this review and we wanted to reply accordingly. We will address your concerns and comments point-by-point.
>
> > 1. Core Claim Undermined by Own Results
>
> The idea here is that original visualizations, by definition, contain all the information in the table. The finding in Table 2 is that having all this information is at least not detrimental to performance, and it helps prepare models to be more robust when exposed to real-world visualizations (as shown in Table 11, which the reviewer correctly highlighted in the strengths).
>
> Examplewise, this experiment uses the same examples as MMTab’s in order to isolate the effect of table representations from other factors such as the number of examples or the instruction phrasing. As we pointed out, this subset is limited in the sense that these examples do not require any of the restored visual information to be answered or solved (It would have been a pleasant surprise to observe some correlation between the missing visual features and the answers to these examples, but the results do not support this, as expected). So, with Table 2 we see that the inputs contain more information (all the information), and still this additional information is neither treated as noise nor detrimental to performance. This is good because we have richer representations available to exploit further and also because (as it turns out) it prepares models to be more robust when such information is present  in real-world pixel-based scenarios.
>
>
> > 2. Circular Argument
>
> Yes, we agree that in the previous version of the paper this may have appeared to be the case, and we thank the reviewer for pointing it out. So for this rebuttal, we created a test set to evaluate exactly this characteristic, and we see that models trained with TABLET significantly outperform others (Table 5 in the revised paper).
>
> The details can also be found in the newly added paragraph (Section 5: TABLET-Bench) in the revised version of the paper. But in essence, we created a human-annotated Visual QA test set to test whether our models, despite being trained on tabular tasks predominantly designed without leveraging visual features of tables, would benefit from their training on TABLET to solve tasks that require combining visual and tabular understanding. We use a visual complexity metric we created for this rebuttal (more on this in Section 5: Visual complexity analysis)  to select a group of tables with a high level of visual complexity and use human annotators to generate a set of 100 QA examples that feature questions that can only be answered by combining attention to visual cues with proficiency in table understanding. Annotators were trained based on a few examples to get the type of questions needed without any contact whatsoever with the models. The resulting questions resemble natural visual question instructions such as "What model is the red car in the table?" or "In what year did the king holding a staff with a red cross die?". We evaluated the 5 models shown in Table 3 and found that models fine-tuned on TABLET outperform the other best model by +7 to +10 points on this benchmark (this difference is statistically significant). We are including this test set in our release, and included this analysis in Section 5 in the revised version of the paper.
>
>
> | **TABLET-Bench** | acc  |
> |-----------------------|---|
> | 0-Shot                | 55 |
> | MMTab                 | 54 |
> | TABLET-S          | 62 |
> | TABLET-M         | **65*** |
> | TABLET-L          | 63 |
>
> > 3. Diluted Conclusion on "Mixed" Data
>
> Yes, this is an interesting point. The additional examples that synthetic tables allow us to incorporate increase the proficiency of models at solving tabular tasks (in this area, more data generally leads to better performance), but they do not help models improve robustness when handling real-world table representations. We can see how in Table 11 (this is the table of the performance downgrade, indexes changed in the revised paper) TABLET-B org’s performance degrades slightly less than mix, and although both remain far better than any other model, the difference in robustness is minimal, the improvement comes at table solving skills.
>
> Why do we include 11% of synthetic images in our dataset? Because they do add value in terms of tabular task solving skills, and for a submission to the dataset track we wanted to provide the best possible resource for the vision table understanding community.
>
> We continue in the next comment...

---

> ### Author Response · Authors · 2025-11-27
>
> > 4. Limited "Real-World" Diversity
>
> This could seem like the case initially, but we can show (and quantify) that there are tables in our dataset that are very far from uniform. In fact, the visual chaos we witnessed when analysing wikipedia tables at the beginning of this work was one of the main motivations that inspired this work. For our rebuttal we wanted to measure this complexity in tables and we created a visual complexity metric. The details of this metric are explained in Appendix 10 in the revised version of the paper but in summary, we created a single metric that aggregates multiple structural and visual table characteristics: column and row span irregularity, color and font diversity, embedded image presence, color entropy, and more, into a single normalized value. This metric is very effective at creating a scale of visual complexity across the 1.4M tables in our dataset (you can find some examples in Figure 18 in the Appendix of the revised paper and the distribution of table scores in Figure 19). We would like to highlight that creating such a metric would have been significantly more difficult without access to both a format-aware serialized representation (HTML) and the rendered visual form that we include in TABLET.
>
>
> We understand however the point of the reviewer with this weakness, and we would like to highlight that we not only incorporate Wikipedia tables (which we have shown can be highly irregular), but also include 27% of PDF document extracted tables from PubTabNet and TabMWP. And while diversity can always be increased (as with any data collection work), our dataset already allows models handling visually complex tables better in several tasks. In the revised paper (Section 5: Visual Complexity Analysis), we added a full analysis on how the performance of models trained with different datasets (or no dataset at all) behaves across different levels of visual complexity and across tasks.
>
> We hope this addresses the reviewer’s concerns regarding the contribution of TABLET as a resource for the visual table understanding community, as well as the value of the newly added analysis and experiments. We would very much appreciate it if you would take this into account when reevaluating the rating. Please, do not hesitate to write further comments, should you have more concerns.

---

### Official Review · Reviewer_P6aP · 2025-11-02

**Soundness:** 3
**Presentation:** 3
**Contribution:** 2
**Rating:** 4
**Confidence:** 3

**Summary:**

The paper introduces TABLET, a large-scale dataset for visual table understanding (VTU). Compared to existing resources (e.g., MMTab), TABLET (i) reconstructs tables in their realistic visual form from web/Wikipedia sources, (ii) unifies them into an image + HTML + metadata format, and (iii) covers 20 VTU-related tasks. The authors fine-tune Qwen2.5-VL-7B on this data and show improvements on both seen and unseen VTU tasks. The main value of the paper is on the data/engineering side and on providing a more realistic training distribution.

**Strengths:**

1.The motivation is real: current VTU data is mostly synthetic or visually simplified, which causes a distribution gap for vision-language models that must operate on real web-like tables.

2.The dataset is large, diverse, and traceable back to the original sources, which is helpful for reproducibility and later task extension.

3.Experiments are fairly comprehensive (synthetic vs. real-style vs. mixed) and the results are consistent with the stated motivation.

**Weaknesses:**

1. The paper does not introduce a new learning objective, a new model architecture, or a principled framework for unifying VTU tasks. The core message is essentially “better/realistic data → better performance,” which is valuable but closer to a dataset/empirical paper than to a method/theory paper.

2. The paper repeatedly argues that preserving realistic visual styles leads to better generalization, but the current evaluations mostly show dataset-level gains, not an analysis of which visual factors (borders, multi-row headers, fonts, layout noise) the model actually exploits.

**Questions:**

None

---

> ### Author Response · Authors · 2025-11-26
>
> We sincerely thank the reviewer for taking the time to read and review our paper, and especially for their feedback which led us to dive deeper into the characteristics of our dataset and introduce a visual complexity metric that we think improves the overall contribution of the paper. We hope to address your concerns in the following response.
>
> > 1. The paper does not introduce a new learning objective, a new model architecture, or a principled framework for unifying VTU tasks. The core message is essentially “better/realistic data → better performance,” which is valuable but closer to a dataset/empirical paper than to a method/theory paper.
>
> We present a resource that allows tackling a problem that could not be addressed before due to the lack of such resources. While we agree with the reviewer that new architectural changes are very interesting, it is very difficult to develop a technique for a problem like this without a dataset like this. The data-driven solution we present in Section 4 was only possible with the dataset we are presenting. The model we are releasing outperforms other models in 10 out of 14 benchmarks thanks to our dataset. While simple in its architectural execution, the complexity of achieving this model relies on the data we are releasing, which we believe is a solid contribution for the datasets and benchmarks track to which we are submitting.
>
> Beyond the direct benefits of the data we have gathered, our dataset also provides complementary metadata that is not present in other datasets (cross-dataset example traces, pairs of visualizations and original HTML representations, etc.) and allows the development of new techniques like the one we developed during this rebuttal period, which we explain in our response to the second weakness.
>
> > 2. The paper repeatedly argues that preserving realistic visual styles leads to better generalization, but the current evaluations mostly show dataset-level gains, not an analysis of which visual factors (borders, multi-row headers, fonts, layout noise) the model actually exploits.
>
> Thank you for this line of feedback. We agree with the reviewer that an analysis like this was lacking, so we have conducted an analysis on how different visual factors affect models performance and how models trained on our dataset behave under different levels of visual complexity. We included this analysis in the “Visual complexity analysis” paragraph in Section 5.
>
> First, we analyzed the performance of the models with different magnitudes of certain format and visual characteristics, such as the number of embedded images or the number of different text and background colors present in the table. We found that analyzing performance differences in individual components was too simplistic, as tables with, for example, a large number of embedded images could end up looking more regular and visually simple due to low diversity in other components (such as structure or cell color diversity). Visual complexity of a table is a multi-dimensional characteristic resulting from a combination of factors and requires a metric that combines the influence of all these factors into one.
>
> Thus, we have created a single metric that aggregates multiple structural and visual table characteristics: column and row span irregularity, color and font diversity, embedded image presence, color entropy, and more, into a single normalized value (please find the full details of this metric in Appendix 10 of our new version of the paper). This metric is very effective at creating a scale of visual complexity across the 1.4M tables in our dataset (please see Figure 19 in the Appendix of the revised paper). We would like to highlight that creating such a metric would have been significantly more difficult without access to both a format-aware serialized representation (HTML) and the rendered visual form that we include in TABLET.
>
> We compute this score for all tables in TABLET and evaluate each model's performance within each group for each benchmark. Figure 2 in the revised paper shows results for the tasks where model differences are most pronounced (see Appendix 11 for results across all benchmarks).
>
> Across these tasks, models trained on our dataset remain robust even as tables deviate substantially from regular, visually simple layouts. The analysis also shows that in some tasks like TABMWP, fine-tuning on table images is crucial to avoid performance degradation, and TABLET trained models consistently show higher performance overall.
>
> We believe this is a very interesting analysis and we thank the reviewer again for leading us to perform it.
>
> We continue in the next comment.

---

> ### Author Response · Authors · 2025-11-26
>
> In addition to this analysis, during this rebuttal period we had the opportunity to show that the benefits of our dataset also extend to different training procedures and model families. We performed SFT using LoRA on Gemma 3-4B and obtained similar benefits as with the fully fine-tuned Qwen2.5-VL-7B backbone. In our revised version of the paper, we report how Gemma 3 fine-tuned on TABLET-M (the fine-tuning is still ongoing, and we report the state of this model at 75% of one epoch. We will update the final results when training finishes, but the performance trend is positive) shows that even at 75% of an epoch, the model trained with TABLET-M significantly outperforms the same model fine-tuned on MMTab across all benchmarks. Thus, we see that the benefits of our dataset also transfer to other training procedures and model families. More details on this experiment can be seen in the last paragraph of Section 5 in the newly revised paper.
>
> We would also like to highlight that during the rebuttal period, we had the opportunity to create a benchmark of the main characteristic of Visual Table Understanding that motivated our work. We created a human-annotated Visual QA test set to test whether our models, despite being trained on tabular tasks predominantly designed without leveraging visual features of tables, would benefit from their training on TABLET to solve tasks that do require combining visual and tabular understanding. We use our visual complexity metric to select a group of tables with a high level of visual complexity and use human annotators to generate a set of 100 QA examples that feature questions that can only be answered by combining attention to visual cues with proficiency in table understanding. These questions resemble natural visual question instructions such as "What model is the red car in the table?" or "In what year did the king holding a staff with a red cross die?" We see that models fine-tuned on TABLET outperform the other best model by +7 to +10 points on this benchmark. We are including this test set in our release, and more information can be seen in the TABLET-Bench paragraph of Section 5 in our revised paper.
>
> We apologize for the rather late response as we have been busy working on applying the feedback we have gathered during the rebuttal. We hope we addressed the reviewer's concerns regarding the contribution of TABLET as a resource for the visual table understanding community, along with the value of the newly added analysis and other experiments. We would very much appreciate it if you would consider it when reevaluating the rating.

---

### Official Review · Reviewer_czgq · 2025-11-03

**Soundness:** 2
**Presentation:** 3
**Contribution:** 2
**Rating:** 2
**Confidence:** 5

**Summary:**

This paper presents TABLET, a large-scale dataset for UTV, containing 4M examples. The key contribution is the collected TABLET from sources from Wikipedia and PubTabNet. The dataset are split into train, dev, and test sets and comprising comprehensive table-related tasks. Experiments with Qwen2.5-VL-7B show good performance gains on several benchmarks, and the authors argue that TABLET improves robustness on real-world table images.

**Strengths:**

+It collects a large-scale dataset combining multiple VTU sources.
+It provides both HTML and image representations with metadata, which facilitate the document understanding community.
+Experiments on the Qwen2.5-VL models demonstrating the effectiveness of the collected dataset.

**Weaknesses:**

1. The author presents a new dataset, TABLET, and the author claims that the dataset contains 4M examples. However, nearly 30% are existing, the author should describe this more rigorously.
2. No new model, algorithm, or theoretical insight in the paper. The paper mainly merges existing datasets and collected new table data on the network.
3. One of the author’s main motivation is that synthetic data are of lower quality than real-world data. However, the majority of the dataset they collected still consists of screenshot-based images, many of which are taken from sources like Wikipedia. This seems somewhat inconsistent with the author’s stated motivation?
4. Poor experiments. In Tab.2 and 3, the performance improvement by the proposed dataset appears to be limited.
5. What confuses me is that the paper only reports results for qwen and the results fine-tuned on qwen, without any comparison to other models. This is completely unreasonable for a research paper.

**Questions:**

Please refer to the weaknesses section above.
It seems that the author merely proposed a dataset and fine-tuned qwen with it, without providing comparisons to other models. I believe the paper still needs further polishing.

---

> ### Author Response · Authors · 2025-11-26
>
> We thank the reviewer for their comments and suggestions, and especially for taking the time to read and review our paper. In the following response, we will address their concerns and comments point-by-point.
>
> > 1. The author presents a new dataset, TABLET, and the author claims that the dataset contains 4M examples. However, nearly 30% are existing, the author should describe this more rigorously.
>
> We appreciate the reviewer's feedback regarding our dataset composition. In the revised version of our paper (line 210), we clarify further that we include tables from PubTabNet and TabMWP because our goal is to emphasize original visualizations, and these datasets fortunately already contained tables in their original representation extracted from documents. However, we create and incorporate new HTML representations of these tables to enable analyses such as the visual complexity metric we developed during the rebuttal period (more on this when we address the stated weaknesses).
>
> > 2. No new model, algorithm, or theoretical insight in the paper. The paper mainly merges existing datasets and collected new table data on the network.
>
> We would like to emphasize that this work was submitted to the datasets and benchmarks track, where the primary contribution is the resource itself rather than novel algorithms or architectures. As we state in line 316: "Exploring the best backbone model for VTU or advanced SFT strategies is outside the scope of this work."
>
> Our main contribution lies in the collection, unification, and tracing of original table visualizations as they appeared at the time of crawling, which existing datasets do not provide. Having these visual representations is crucial to make models robust to real-world table visualizations as we show in Table 3, Table 11, the newly incorporated visual complexity analysis in Section 5, and the new Visual QA test results in Table 5.
>
> We use consistent SFT configurations across all experiments to maintain fair comparison between datasets (not models), as our goal is to evaluate the value of the data itself. However, and despite this simple training approach, we release a model that outperforms other models on 10 out of 14 benchmarks, demonstrating the quality and utility of our resource.
>
> > 3. One of the author’s main motivation is that synthetic data are of lower quality than real-world data. However, the majority of the dataset they collected still consists of screenshot-based images, many of which are taken from sources like Wikipedia. This seems somewhat inconsistent with the author’s stated motivation?
>
> We would like to clarify our position on real-world visual representation. Our screenshots represent tables exactly as they would appear in real-world scenarios, the way a pixel-based agent or any VLM would encounter them in practice. The key difference is that rendering tables from existing serialized data (as done in synthetic datasets) produces visually simplified representations that deviate substantially from how these tables actually appear in the wild. Our screenshots preserve all visual complexity, styling, fonts, colors, and layout irregularities that models must handle in deployment. This is fundamentally different from synthetic rendering, which strips away these real-world visual characteristics.
>
> To directly address this concern and demonstrate the value of training on these original visual representations, we have created a human-annotated Visual QA test set specifically designed to evaluate whether models can leverage visual cues in table understanding. We select tables with high levels of visual complexity and use human annotators to generate a set of 100 question-answer examples featuring questions that can only be answered by combining attention to visual cues with proficiency in table understanding. These questions resemble natural visual question instructions such as "What model is the red car in the table?" or "In what year did the king holding a staff with a red cross die?". Evaluations on this test set show that models fine-tuned on TABLET significantly outperform the others by +7 to +10 points on this benchmark (full results in Table 5 and in this response for convenience). We are including this test set in our release, and more information can be seen in the TABLET-Bench paragraph of Section 5 in our revised paper.
>
> | **TABLET-Bench** | acc  |
> |-----------------------|---|
> | 0-Shot                | 55 |
> | MMTab                 | 54 |
> | TABLET-S          | 62 |
> | TABLET-M         | **65*** |
> | TABLET-L          | 63 |
>
>
> We continue in the next comment.

---

> ### Author Response · Authors · 2025-11-26
>
> > 4. Poor experiments. In Tab.2 and 3, the performance improvement by the proposed dataset appears to be limited.
>
> We want to highlight the fact that we are starting from a very strong baseline, Qwen2.5-VL is already highly proficient in VTU tasks. And despite this challenging baseline, we achieve: (1) Statistically significant improvements of +14.3, +3.4, +4.1, +26.0, +5.6, +24.5, +22.4, +14.2, +0.2, and +1.8 points respectively (all marked with * indicating statistical significance) with respect to the best score of any other model. These results using only a simple 3-epoch SFT procedure maintained for fair comparison.
>
> The fact that we can significantly outperform such a strong baseline with a straightforward training procedure demonstrates the substantial value of the resource itself.
>
> However, during this rebuttal period we also conducted additional experiments on Gemma 3-4B using a different training procedure (LoRA-based SFT) to verify that the benefits of our dataset transfer to different model families and training methodologies. In our revised paper (last paragraph of Section 5), we report that even at 75% of one epoch, the Gemma 3 model trained with TABLET-M consistently outperforms the same model fine-tuned on MMTab across all benchmarks. The training is still ongoing, and we will update the final results when it finishes, but the performance trend is positive (as we also report results for evaluation on training on a 25% of an epoch). These results show that the benefits of our dataset also transfer to other training procedures and model families. More details on this experiment can be seen in the last paragraph of Section 5 in the newly revised paper.
>
>
> > 5. What confuses me is that the paper only reports results for qwen and the results fine-tuned on qwen, without any comparison to other models. This is completely unreasonable for a research paper.
>
> We do report results for other models in Appendix 8.4 (referenced in line 318), we include them here for convenience:
> | Model                                      | WikiBio | ToTTo | WikiTQ | TabMWP | HiTab | TabFact | FeTaQA | TAT-QA |
> |--------------------------------------------|---------|-------|--------|--------|-------|---------|--------|--------|
> | Table-LLaVA 7B (Zheng et al., 2024b)       | 5.7     | **16.3** | 17.7   | 47.5   | 10.6  | 53.6    | 7.5    | 1.8    |
> | Qwen2.5-VL 7B (Qwen Team, 2025)            | 2.5     | 9.1   | **53.4** | 59.1   | 31.2  | 73.9    | 7.0    | 6.9    |
> | DocOwl2 8B (Hu et al., 2025)               | 1.8     | 1.2   | 18.9   | 0.5    | 9.5   | 54.3    | 9.6    | 2.9    |
> | InternVL3 8B (Zhu et al., 2025)            | 4.6     | 11.9  | 45.3   | 83.9   | 40.8  | 64.3    | 3.8    | **13.0** |
> | InternVL3 14B (Zhu et al., 2025)           | **6.3** | 12.4  | 51.6   | **86.8** | **52.0** | **75.1** | 6.6    | 11.9   |
> | Gemma-3 12B (Team et al., 2025)            | 5.1     | 5.4   | 39.1   | 72.3   | 24.2  | 69.5    | **20.3** | 2.5    |
>
> We acknowledge that this reference could be more prominent, and we included a reference to it in the caption of Table 3. Additionally, as mentioned in the previous weakness response, during the rebuttal period we successfully fine-tuned Gemma 3-4B using LoRA on TABLET-M, showing that the benefits of our dataset transfer across different training procedures and model families (see the last paragraph of Section 5 in our revised paper).
>
>
> > Please refer to the weaknesses section above. It seems that the author merely proposed a dataset and fine-tuned qwen with it, without providing comparisons to other models. I believe the paper still needs further polishing.
>
> We would like to emphasize that we have developed a data-driven solution that resulted in a model that outperforms other models in 10 out of 14 benchmarks. While simple in its architectural execution, the complexity of achieving this model relies on the data we are releasing, which we believe is a solid contribution for the datasets and benchmarks track to which we are submitting.
>
> We apologize for the rather late response as we have been busy working on applying the feedback we have gathered during the rebuttal. We hope we addressed the reviewer's concerns regarding the contribution of TABLET as a resource for the visual table understanding community, along with the value of the newly added changes. We would very much appreciate it if you would consider it when reevaluating the rating.

---

### Author Response · Authors · 2025-11-28
**Updated Results: Gemma 3-4B LoRA Fine-tuning (1 full epoch)**

In addition to the additions we already mentioned in individual responses to reviewers (Visual Complexity Analysis and Visual QA test set evaluation), we have updated the results for Gemma 3 LoRA fine-tuning on MMTab and TABLET-M with results from our model fine-tuned on 1 full epoch (compared to the 1/3 and 3/4 of an epoch we reported in our initial response). Following what we expected from our preliminary results reported in the revision we submitted during our review response, Gemma 3-4B trained on 1 epoch of TABLET-M outperforms on 6 out of 8 benchmarks. Confirming that the benefits of our dataset also extend to this training procedure and model family.

| Model | WikiBio | ToTTo | WikiTQ | TabMWP | Hitab | TabFact | FeTaQA | TAT-QA |
|-------|---------|-------|--------|--------|-------|---------|--------|--------|
| 0-Shot | 4.1 | 4.9 | **35.1*** | 65.2 | 20.6 | **62.1*** | 20.5 | 7.9 |
| MMTab | 8.9 | 12.7 | 30.3 | 69.3 | 32.0 | 53.3 | 8.4 | 4.3 |
| TABLET-M | **12.7*** | **24.7*** | 29.6 | **71.7*** | **35.3*** | 57.9 | **30.9*** | **12.3** |

Best results in bold, * indicates statistically significant difference (p < 0.05, using bootstrap resampling).

---

### Author Response · Authors · 2025-12-02
**Summary of Contributions During the Rebuttal Period**

We thank all reviewers for their time and the feedback provided. During the rebuttal period, we addressed the key concerns raised and incorporated three major additions:
1. To test whether TABLET’s benefits extend beyond Qwen2.5-VL and a single training setup, we fine-tuned Gemma 3 - 4B using LoRA on TABLET-M and MMTab. When evaluating the resulting models, we see that TABLET remains the best Visual Table Understanding (VTU) dataset for improving performance across VTU tasks, confirming our original findings with Qwen2.5-VL-7B and showing that TABLET’s value is not tied to a single model or training procedure. We include the results of this analysis in the revised version of our paper.

| Model | WikiBio | ToTTo | WikiTQ | TabMWP | Hitab | TabFact | FeTaQA | TAT-QA |
|-------|---------|-------|--------|--------|-------|---------|--------|--------|
| 0-Shot | 4.1 | 4.9 | **35.1*** | 65.2 | 20.6 | **62.1*** | 20.5 | 7.9 |
| MMTab | 8.9 | 12.7 | 30.3 | 69.3 | 32.0 | 53.3 | 8.4 | 4.3 |
| TABLET-M | **12.7*** | **24.7*** | 29.6 | **71.7*** | **35.3*** | 57.9 | **30.9*** | **12.3** |

Best results in bold, * indicates statistically significant difference (p < 0.05, using bootstrap resampling).

2. We analyzed how models trained with TABLET and other VTU datasets perform across different levels of table visual complexity. In tasks where other models’ performance degrades as complexity increases, TABLET-trained models remain notably more robust, showing that TABLET meaningfully improves robustness to real-world table visual variability in several VTU tasks. We included this analysis in Section 5 of the revised paper.
To perform this analysis we used TABLET’s paired original HTML and visual table representations to develop a new visual complexity metric that aggregates multiple structural and visual characteristics (row/column span irregularity, color/font diversity, embedded images, color entropy) into a single normalized score. This metric allowed us to separate examples from regular, minimally formatted tables to tables with richer layouts that rely on stylistic cues, images, and irregular structures to convey information. Examples of tables at different scores can be found in Figure 18 and full details of this metric are explained in Appendix 10 in the revised version of the paper.

3. We created a human-annotated Visual QA test set to validate whether TABLET’s visually lossless representations allow models to better solve tasks that require both visual and textual understanding of tables. Models trained on TABLET outperform all baselines by +7 to +10 points on this benchmark, showing that training on TABLET’s original visualizations improves their visual table-understanding skills. This test set and its evaluation are included in Section 5 of the revised paper and will be released alongside the dataset.
| **TABLET-Bench** | acc  |
|-----------------------|---|
| 0-Shot                | 55 |
| MMTab                 | 54 |
| TABLET-S          | 62 |
| TABLET-M         | **65*** |
| TABLET-L          | 63 |

Best results in bold, * indicates statistically significant difference (p < 0.05, using bootstrap resampling) between the marked value and the highest of the other group (in this case Qwenn2.5-VL on 0-shot.

With these additional experiments we demonstrate that TABLET (1) improves VTU performance across model families and training procedures, (2) improves robustness to real-world table visual variability in the tasks we studied, and (3) improves models’ ability to solve Visual Table Question Answering. We hope the Area Chair will consider the value of this dataset as a resource for the multi-modal and Table Understanding community as well as the improvements incorporated during this rebuttal period.

---

### Meta-Review · Area_Chair_mZiF · 2026-01-07

**Summary:**

This paper introduces TABLET, a large-scale dataset for visual table understanding that preserves original visual table renderings and provides paired image - HTML representations with rich metadata and provenance. Reviewers generally agree that the paper identifies an important train–test mismatch in VTU benchmarks and that the dataset constitutes a substantial engineering and resource contribution.

Concerns were raised regarding the lack of methodological novelty, the initially limited evidence that visual fidelity directly improves task performance, and the concentration of sources (e.g., Wikipedia). During rebuttal, the authors added cross-model validation, a visual complexity analysis, and a human-annotated visual QA benchmark, which together strengthen the empirical case that TABLET improves robustness and visual reasoning beyond existing datasets.

**Reviewer Concerns:**

Addressed:

- Generalization beyond a single backbone was addressed via additional experiments on Gemma 3 with a different training procedure.

- The concern that visual fidelity was not directly validated was mitigated by the newly introduced visual complexity analysis and the human-annotated visual QA test set.

- Clarifications were provided on dataset composition, reuse of existing sources, and the motivation within the datasets & benchmarks track.

Partially Addressed / Remaining:

- The work remains primarily a dataset contribution without new learning objectives or architectures.

- While robustness benefits are clearer, the exact visual factors exploited by models are still only analyzed at an aggregate level.

- Source diversity beyond Wikipedia, while discussed, remains a potential limitation for future extensions.

**Reviewer Scores:**

- Reviewer czgq
Despite additional experiments and clarifications in the rebuttal, this reviewer’s core concerns about novelty and experimental scope are unlikely to change. The score would likely increase to 4.

- Reviewer P6aP
The added visual complexity analysis and robustness evaluation address the main concerns. This reviewer would likely revise the score slightly upward, closer to the acceptance threshold.

- Reviewer WHyK
The new visual QA benchmark and clarified robustness results partially address the reviewer’s concerns. A modest positive adjustment to the score is likely, though some reservations may remain.

- Reviewer qfkf
This reviewer was already positive. The rebuttal mainly reinforces their assessment, and the score would likely remain unchanged.

---

### Decision · Program_Chairs · 2026-01-26

Accept (Poster)